# Tuning lower dimensional superconductivity with hybridization at a superconducting-semiconducting interface

Anand Kamlapure [1], Manuel Simonato[1], Emil Sierda[1], Manuel Steinbrecher [1], Umut Kamber[1], Elze J. Knol[1], Peter Krogstrup[2], Mikhail I. Katsnelson [1], Malte Rösner [1✉] & Alexander Ako Khajetoorians [1✉]

The influence of interface electronic structure is vital to control lower dimensional superconductivity and its applications to gated superconducting electronics, and superconducting layered heterostructures. Lower dimensional superconductors are typically synthesized on insulating substrates to reduce interfacial driven effects that destroy superconductivity and delocalize the confined wavefunction. Here, we demonstrate that the hybrid electronic structure formed at the interface between a lead film and a semiconducting and highly anisotropic black phosphorus substrate significantly renormalizes the superconductivity in the lead film. Using ultra-low temperature scanning tunneling microscopy and spectroscopy, we characterize the renormalization of lead's quantum well states, its superconducting gap, and its vortex structure which show strong anisotropic characteristics. Density functional theory calculations confirm that the renormalization of superconductivity is driven by hybridization at the interface which modifies the confinement potential and imprints the anisotropic characteristics of the semiconductor substrate on selected regions of the Fermi surface of lead. Using an analytical model, we link the modulated superconductivity to an anisotropy that selectively tunes the superconducting order parameter in reciprocal space. These results illustrate that interfacial hybridization can be used to tune superconductivity in quantum technologies based on lower dimensional superconducting electronics.

[1] Institute for Molecules and Materials, Radboud University, 6525 AJ Nijmegen, The Netherlands. [2] Center for Quantum Devices, Niels Bohr Institute, University of Copenhagen, 2100 Copenhagen, Denmark. ✉email: m.roesner@science.ru.nl; a.khajetoorians@science.ru.nl

Hybrid superconductor-semiconductor heterostructures are a pathway toward tailoring superconductivity as well as for creating a platform for topological quantum computing[1,2]. While the changes in electronic structure created by coupling the bulk bands of a superconductor and a semiconductor can modify both the electronic properties in the semiconductor[3,4], as well as the properties of the superconductor, the interface itself can also introduce a new electronic degree of freedom that is often neglected. Toward this end, it has been shown that superconductivity can be enhanced at the interface between dissimilar materials[5–7]. For example, monolayer of iron selenide (FeSe) grown on strontium titanate, a large bandgap semiconductor, exhibits high temperature superconductivity[5], with a critical transition temperature nearly an order of magnitude larger than in the FeSe bulk compound[8]. Likewise, the interface at lanthanum aluminate on strontium titanate exhibits superconductivity in a 2D electron gas residing at the interface between the two insulating compounds[6]. While these examples clearly show that superconductivity can be strongly enhanced by an interface, the mechanism for superconductivity in these heterostructures is still strongly debated making it challenging to understand the essential phenomena responsible for these changes[7,9].

Superconductor-semiconductor heterostructures, derived from elemental superconductors, provide a tunable platform toward quantifying the various influences of the heterostructure on superconductivity[10,11]. So far, the emphasis of numerous studies has been in establishing the interplay between dimensionality and superconductivity, when compared to their bulk counterparts[12–16]. For example, it has been shown that quantum confinement can modulate $T_c$, through quantum well states, leading to the demonstration of 2D superconductivity[12–14]. These approaches rely on supporting semiconductors with a sizeable electronic band gap, which largely suppress unwanted hybridization from the semiconductor that otherwise may lead to quasiparticle poisoning in the superconductor. With the growing interest in quantum technologies based on superconductor-semiconductor heterostructures[1,17], it remains unknown how hybrid band structures created at the junction of two materials can be used to controllably tune superconductivity in lower dimensional structures.

Here, we demonstarted the formation of a hybrid superconductor formed by the interface of a superconducting ultra-thin film of lead and semiconducting black phosphorus. The hybrid band structure induced by the interface renormalized the superconductivity of the lead layers yielding strongly anisotropic characteristics that depend on layer thickness. Using ultra-low

temperature scanning tunneling microscopy (STM) and spectroscopy (STS) down to mK temperatures, we quantified the superconducting gap, which shows an unexpected anisotropic structure defined by a strong non-thermal broadening of the gap, as a function of thickness of the lead film. Using spatial imaging in magnetic field, we quantify the resultant Abrikosov lattice which is derived from strongly anisotropic vortices. We observed concomitantly the presence of a strong moiré lattice, which allows us to ascertain the structure of the film, including the induced strain driven by the interface. Using this determined structure as input, we performed ab initio density functional theory calculations based on a minimal moiré structure and observed that the black phosphorus bands influence both the quantum well states of the lead film around Γ as well as its Fermi surface deeper within the Brillouin zone. The hybrid electronic structure leads to spectral fingerprints that can be seen from both calculations and STS measurements, that strongly differ from signatures of quantum well states (QWS) of nearly free-standing Pb films[18]. We developed a hybrid two-band superconducting model which illustrates that superconductivity can be sculpted across regions of the Fermi surface by considering weak hybridization of the lead bands, including the quantum well sub-bands, with the anisotropic black phosphorus bands. This enabled us to fit and discern the detailed gap structure measured, and to relate the non-thermal broadening of the gap to a $k$-space dependent gap function and an accompanying weighting function that together describe the superconducting quasiparticles. Based on these results, we observed a gradual weakening of the interfacial effect in thicker films, allowing us to distinguish between an ultra-thin limit where the hybrid superconductivity is strongly driven by the interfacial hybridization from a thick film limit where the interface is a weaker perturbation in comparison.

## Results

Black phosphorus (BP) is a narrow band gap semiconductor ($E_G \approx 0.3$ eV) that exhibits a strongly anisotropic electronic dispersion[19–22]. BP cleaves with an atomically flat surface over macroscopic length scales, exposing the (001) surface. Closed Pb(111) films grow epitaxially on the clean surface of cleaved BP when utilizing a two-stage growth process (see methods for two-stage growth) as depicted in Fig. 1a. Pb grown at room temperature leads to island growth without a wetting layer, and we observed a conductance gap which limits the ability to quantify the superconducting gap when the film is not closed (see Supplementary Fig. 1). Using constant-current STM imaging (Fig. 1b), we observed that the absolute thickness in

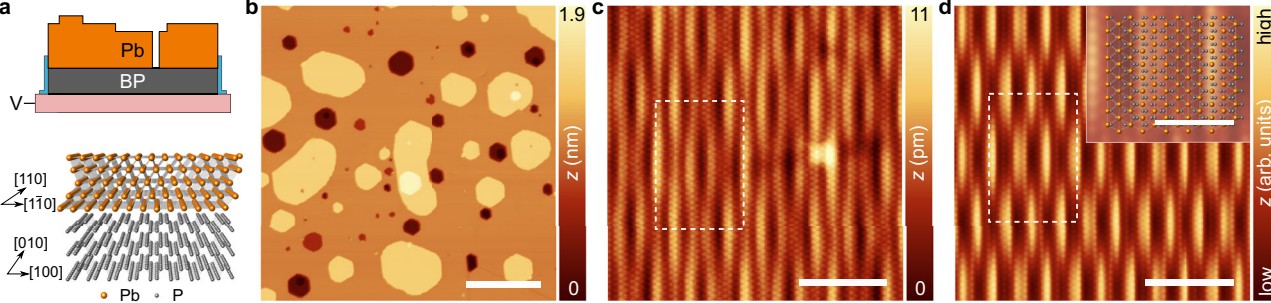

**Fig. 1 Hybrid superconducting heterostructure of lead and black phosphorus. a** Top: Sketch of the sample geometry showing the cross section of the ultra-thin Pb(111) film and the BP crystal. The blue patches on the two sides represent side contacts. Bottom: Crystal structure of Pb and BP showing the relative orientation between the Pb lattice and the BP lattice. **b** Constant-current STM image of a 9 ML Pb film ($V_s = 600$ mV, $I_t = 10$ pA, $T = 1.3$ K, scale bar = 50 nm). **c** Atomically resolved constant-current STM image measured on a 7 ML Pb film showing the persistent moiré structure between Pb and BP. The white dashed rectangle represents moiré supercell. ($V_s = 100$ mV, $I_t = 100$ pA, $T = 30$ mK, scale bar = 5 nm). **d** Simulated moiré structure as described in Supplementary Note 1. The white dashed rectangle represents moiré supercell (scale bar = 5 nm). Inset: lattice structure of the interface overlaid on the zoomed-in image of the simulated moiré structure (inset scale bar = 2 nm).

monolayers ($N$) of a given film can be identified using holes in the film which penetrate directly to the BP surface (Supplementary Figs. 2 and 3a). This can be combined with the unique layer-dependent spectroscopic fingerprinting of the local density of states (LDOS), which is discussed later. The typical Pb film exhibits monolayer variations in absolute height, with regions from 7 to 11 monolayers (ML) illustrated in Supplementary Fig. 2. The presence of monolayer variations in height is in stark contrast to the expected bilayer height variations driven by the quantum well states common to the growth of Pb films on numerous other surfaces[23–28]. The quenching of the quantum size effect in these Pb films provides a first indication of a hybrid electronic structure that is different than the well-studied nearly free-standing Pb films[18].

High-resolution constant-current imaging on various film thicknesses reveals a persistent and strong moiré lattice (Fig. 1c) in addition to atomic resolution of the Pb(111) film with the expected symmetry. The moiré lattice exhibits a rectangular unit cell (5.03 nm × 7.35 nm) which extends along the zig-zag direction of the BP [010] rows. We observed the same moiré pattern, regardless of sample and film thickness, from 3 to 30 ML. We simulated the moiré lattice considering both the interference between the Pb and BP atomic lattices, as well as the strain induced in the Pb film (see Supplementary Note 1 and Supplementary Fig. 4). Based on this, we reproduced the experimentally observed images (Fig. 1d), which confirms the atomic structure of the Pb film relative to the underlying BP. Based on these simulations, we find that all observed films exhibit a uniaxial strain (compressed) of ~1% along the [110] direction of the Pb lattice. As we do not observe a wetting layer and all the films show islands and vacancy islands, the strain is likely relieved in the [111] direction of the Pb.

**Characterization of the superconducting gap**. We quantified the superconducting properties of these Pb films as a function of thickness, using STS down to $T = 30$ mK (Fig. 2a). Spectra measured at various locations on a given terrace reveal a uniform and single superconducting gap of roughly $\Delta \approx 1.29$ meV, which is different than the two-gap structure seen for bulk Pb(111) with $\Delta_1 \approx 1.27$ meV and $\Delta_2 \approx 1.42$ meV[29,30]. The fully gapped spectra for 7–9 ML are shown in Fig. 2a and can be qualitatively characterized by strongly broadened gap structures, which cannot be attributed to thermal broadening as we detail below. To illustrate the observed gap renormalization, we compared the superconducting gap of Pb on BP to Pb films grown on Si(111), which were measured at similar conditions. For the case of Pb on BP, in the limit of the thinnest films, we observed a significant renormalization of the gap structure in comparison to Pb on Si(111) which can be fitted by conventional methods considering a U-shaped gap.

The overall modifications to the gap structure driven by the interface, can be described by three signatures each with different relative contributions: (i) a broadening of the conductance near the coherence peaks, (ii) a suppression of the coherence peaks and (iii) a gradual and rounded onset of conductance at the gap edge. These various observations cannot be explained by temperature broadening of a BCS gap, which will evenly broaden the total gap structure. Also, the gap cannot be accurately fitted with conventional formulas based on BCS theory[31,32]. Therefore, the most natural consideration would be to account for an induced anisotropic gap due to the BP interface, namely an angle-dependent superconducting gap, resulting from a renormalization of the Fermi surface from circular to ellipsoidal[33,34]. However, considering solely an anisotropic gap function, at sufficiently low temperature together with an ellipsoidal Fermi surface, should lead to a nontrivial modification to the gap structure (e.g. kinks)

that should be observable at our experimental temperatures[34]. This was not observed in the temperature-dependent experimental spectra and thus motivates additionally considering a $k$-dependent weighting of the gap function, as we discuss later. Non-thermal broadening has been seen in point contact spectroscopy measurements in Pb/InAs tunnel junctions, however, such effects do not stem from the interface but rather from the junction properties[35]. Unlike the expected bilayer oscillation in $\Delta(N)$ seen for Pb/Si[12–14], the overall gap structure and $\Delta(N)$ do not show any pronounced changes or layer-dependent oscillations and remain robust for a given film. While the variations in the gap are small for a given sample, we observed strong variations between samples with films of similar thicknesses, which we discuss later. Likewise, we did not observe strong spatial variations in the superconducting gap depending on the probed location with respect to the underlying moiré lattice. We also did not observe any proximity induced gap in the BP, which could be directly measured through holes in a given Pb film (Supplementary Fig. 3). Pb leads to an $n$-doping of the BP near the interface, pushing the Fermi energy into the intrinsic bulk band gap of the BP and thus this doping quenches any carriers in the BP that can be proximitized. This is in contrast to Pb grown on other narrow band gap semiconductors, such as $Bi_2Te_3$, where the samples are intrinsically $n$-doped and exhibit a sizeable proximity effect[36]. Finally, we observed a relative difference in the gap structure for ultra-thin films ($< \approx 12$ ML), compared to films approximately 30 ML thick. For the latter, we observed gap structure that still exhibit a degree of anisotropy, but more closely resemble the gap structure of Pb/Si(111). This suggests the interface electronic structure is responsible for the observed variations.

**Vortex characteriziation**. In order to characterize the renormalization of the superconducting order parameter, we studied spatially resolved maps of the resulting vortices in an applied out of plane magnetic field for as grown 7.3 ML Pb films. For $B_\perp = 50$ mT, we observed a pronounced triangular Abrikosov lattice (Fig. 2b) with an average vortex-vortex distance of $d_v \approx 217$ nm. High-resolution imaging reveals that this vortex lattice is composed of strongly anisotropic vortices (Fig. 2c), which strongly deviate from circular symmetric vortices seen in thin Pb films[37–39]. This change in vortex structure is another indication of a significant renormalization in the order parameter. Furthermore, the vortex structure does not resemble a simple ellipsoidal structure seen for other superconductors described by an anisotropic Fermi surface[33,34]. Simultaneous vortex imaging at the coherence peak in the same area (Supplementary Fig. 5) reveal circularly shaped vortices, indicating a strong $k$-dependent dispersion of Bogoliubov-de Gennes (BdG) quasiparticles. These strongly anisotropic vortices vary slightly in their shape and size, likely due to weak disorder resulting from variations in the film thickness. They exhibit clear deviation from the isotropic vortex structure seen in type-II Pb films[37–39], which can be best described as showing reduced symmetry from the hybrid interface where Pb/BP exhibits a three-fold/two-fold structure respectively, and a modulation due to the moiré structure. As shown in Supplementary Fig. 6, we observed a transition toward nearly isotropic vortex structures in thicker films (~30 ML), which agrees with the evolution toward a more isotropic gap structure for thicker films, seen in Fig. 2. These observations indicate that, in ultra-thin films, the order parameter is strongly influenced by the hybrid structure formed from the BP interface, and that these effects are more effectively screened as the film thickness increases.

By performing spatially dependent spectroscopy along a given vortex, the anisotropic character can be quantified

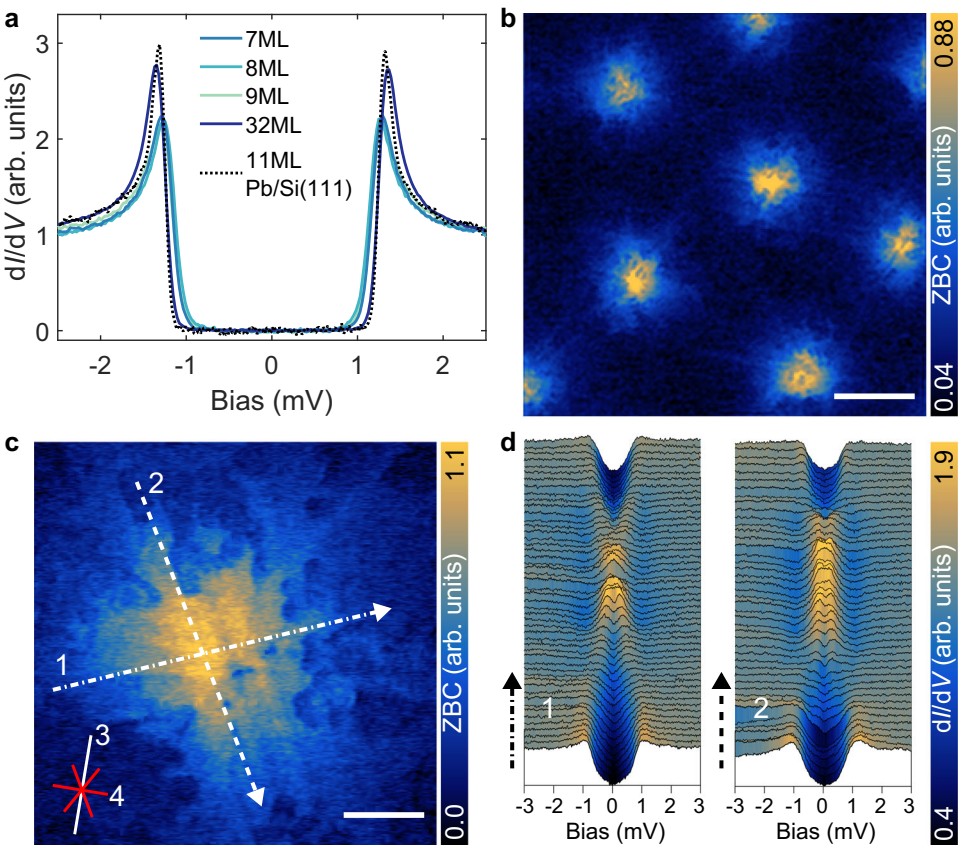

**Fig. 2 Hybrid superconducting gap and anisotropic vortex structure. a** d$I$/d$V$ spectra measured at $T = 30$ mK on different layer thicknesses of the Pb film grown on BP, plotted together with the spectrum measured on 11 ML Pb grown on Si(111). For comparison, all the spectra are normalized at $V = 3$ mV. (Pb/BP: $V_{stab} = 5$ mV, $I_{stab} = 200$ pA, $V_{mod} = 50$ µV; Pb/Si(111): $V_{stab} = 5$ mV, $I_{stab} = 200$ pA, $V_{mod} = 20$ µV). **b** Zero bias conductance (ZBC) map measured on as grown 7.3 ML Pb film at $T = 30$ mK and $B_\perp = 50$ mT showing the Abrikosov vortex lattice (scale bar = 100 nm). Image in **c** shows ZBC map of a single vortex (scale bar = 20 nm). Solid lines in the bottom left indicate in-plane crystallographic directions for BP (white line along zig-zag direction [010] denoted by 3) and Pb (red lines along atomic rows of Pb(111) lattice denoted by 4). Imaging parameters for **b** and **c**: $V_{stab} = 10$ mV, $I_{stab} = 10$ pA, $V_{mod} = 200$ µV, $\Delta z = -80$ pm. **d** d$I$/d$V$ spectra measured across the vortex in two directions (1,2) along the lines through the vortex in **c** each of 84 nm in length. ($V_{stab} = 5$ mV, $I_{stab} = 200$ pA, $V_{mod} = 50$ µV).

(Fig. 2d). The d$I$/d$V$ spectra measured along one of the extended stripes (along [110] of the Pb lattice) shows a small splitting of the zero bias peak away from the center and extends over several tens of nanometers (direction 2) before abruptly merging into the coherence peaks. However, d$I$/d$V$ spectra measured in a nearly perpendicular direction (direction 1) show a splitting of the zero bias peak which smoothly merge into the coherence peaks above 10 nm. We chose these two directions for the given vortex such that the spectra were taken on a flat area with the same thickness, and to avoid the island edges. We note that vortices were observed on all films, while the formation of a clear Abrikosov lattice was only seen on thin films with a sufficiently low defect density. A comparison between a zero bias conductance map and the topography shows no strong correlation. However, a slight distortion of the Abrikosov lattice (especially at $B_\perp = 80$ mT) indicates weak pinning of the vortices (Supplementary Fig. 7). Based on magnetic field dependent spectroscopy measurements, we extract $H_{C2} \approx 280$ mT and a coherence length of $\xi \approx 30$ nm (Supplementary Fig. 7). We note that the value of $\xi$ is extracted from a radially averaged profile over the given vortex. A radially dependent analysis of the vortex would require statistics over more vortices with comparable structure, based on a detailed study of a flat film with one layer thickness on the order of the length scale of the Abrikosov lattice constant.

**Ab initio calculations.** To uncover the role of the electronic structure on the observed anisotropic superconductivity, we performed density functional theory (DFT) calculations of both quasi-free standing Pb films as well as for identical films coupled to the BP substrate (see methods and Supplementary Note 2). The mismatch between the BP and Pb lattice constants results in the formation of a moiré pattern. To proceed with the band structure calculations, we thus need to find a reasonably good commensurate approximation with a periodicity that is not too large. To this end, we used a minimal moiré model laterally consisting of two $\sqrt{3}$ Pb unit cells at the experimentally determined 1% strain (Fig. 1c) on three BP unit cells with three BP layers thickness. Periodic boundary conditions force us to strain the BP substrate unit cells. The positions of the Pb atoms are subsequently relaxed. In Fig. 3a we present the resulting Fermi surfaces of a 5 ML Pb film unfolded to the primitive Pb Brillouin zone. Next to the quantum well states around $\Gamma$ we observe a prominent inner hexagonal Fermi surface. These states result from the same Pb band of mixed $p_z$, $p_x$, $p_y$ character, which can efficiently hybridize with the anisotropic BP states. This hybridization imprints modulations to the hexagonal Fermi surface (Fig. 3b) characterized by a slight and strong inward bending in the $k_y$ and $k_x$ directions, respectively. This underlines the two-fold symmetry imprinting to the Pb states from the BP substrate, which as we describe later, influences the superconductivity of the Pb film.

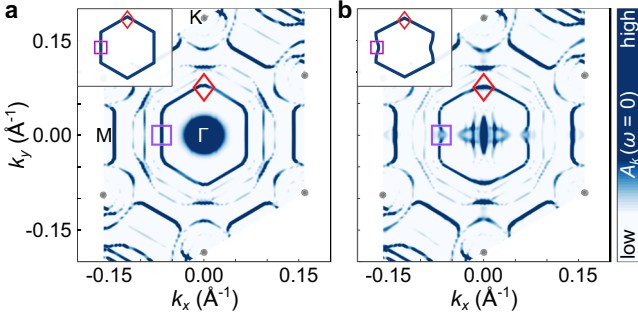

**Fig. 3 Renormalization of the Fermi surface of the hybrid structure.** DFT calculated Fermi surface for 5 ML Pb (**a**) and 5 ML Pb on 3 ML BP (**b**). The insets show a schematic representation of the undistributed inner Fermi surface of 5 ML Pb (**a**) compared to the distorted Fermi surface of 5 ML Pb resulting from the hybridization with underlying BP substrate (**b**). Diamond and square shapes in the inset show the part of the Fermi surface along the directions Γ-K and Γ-M in the corresponding panels.

In order to understand the influence of the hybridization to the quantum confinement, we calculated the resultant band structures of several heterostructures with varying lead thickness. In Fig. 4a, b we show the unfolded results for 5 ML and 6 ML Pb on 3 ML BP. In the $k$-resolved band structures we depict both the full heterostructure and the free-standing Pb film data. This is accompanied by local density of states (LDOS) calculations in which we overlay the full heterostructure LDOS with the individual Pb and BP LDOS. This allows us to disentangle the various features in the STS. In the odd (5) ML Pb case, we can cleary see three well defined QWS states[40] around Γ (labeled a,b,c in Fig. 4a) of free standing Pb in the chosen energy interval, which either strongly overlap and hybridize with BP states (a,b) and significantly broaden in the STS spectra, or which shift in energy (c) upon hybridization with the substrate. In the even (6) ML Pb case we initially find four QWS states (labeled i, ii, iii, iv in Fig. 4b), which again either strongly overlap and hybridize with BP states (i and iii) or shift in energy upon hybridization with the substrate (ii and iv). Likewise, the 3 ML BP around Γ leave their footprints in the full heterostrucutre spectra, especially in the enery windows $[-2.5, -1.5]$ eV, $[0, 0.5]$ eV, and $[0.75, 1]$ eV. The widths and positions of these BP state windows do, however, depend on the BP doping level and the exact number of BP layers in the calculations. In the experimental data, these windows are thus (a) up-shifted due to the intrinsic $p$-doping of BP and are (b) significantly broaded due to the use of thick BP samples as substrate.

To experimentally quantify the electronic structure of the films, we measured d$I$/d$V$ spectroscopy for different film thicknesses (Fig. 4d) and compared this to the calculated LDOS (Fig. 4c). The experimental LDOS exhibits identifiable features reminiscent of the bilayer oscillation of the QWS for Pb films, which are heavily broadened in comparison to the QWS observed for Pb films on larger gap semiconductors[13,41] or surfaces where Pb films are nearly free-standing[18,42]. For example, even layers (4 and 6 ML) show the strongly broadened QWS around $V_S \approx 1.15$ V, which we can identify with the feature iii indicated in the ab initio calculations from Fig. 4b, c. These features are in the experimental STS as well as in the theoretical LDOS data separated via a gap from the features labeled ii and i. In the odd (3, 5, and 7) ML data, we can identify the feature indicated by "b" shown in Fig. 4a and c, which appears in the STS data as a step-like feature between $V_S \approx 0$ and $V_S \approx 1$ V. From the comparison to the ab initio calculations, we conclude that this feature results from the hybridized and overlapping QWS of Pb with the BP

bands. We note that the finer variations seen in the spectroscopy are consistent between sample growths, for films of the same thickness, and do not significantly vary on a given terrace (i.e. do not show strong spatial variations) (Supplementary Fig. 9). These features are also persistent on Pb islands grown at room temperature, enabling a clear spectroscopic fingerprinting of given Pb layers (Supplementary Fig. 10).

In a simple quantum well picture, hybridization of the quantum well boundary with the underlying BP leads to delocalization of the confined wavefunction and weakens the confinement effect. This can yield an additional density of states contribution from the hybrid band structure, in comparison to the expected free-standing QWS. This weakening of the confinement potential may explain the lack of a clear quantum size effect in the grown films. By increasing the film thickness to 28 ML, we reach a regime where the QWS show more expected bilayer oscillation behavior (Supplementary Fig. 11) with some remaining influence from the interface in the form of weak broadening. This observation together with the aforementioned observations regarding the superconductivity in thicker films confirms that by significantly increasing the film thickness, the Pb films still host QWS, which are, however, only weakly perturbed by the BP interface as compared to the ultra-thin limit.

**Hybrid two-band model.** The Bogoliubov-de Gennes (BdG) quasiparticles will be significantly affected by the hybridization of the two subsystems, especially around the Fermi surface in the regions of the Brillouin zone where the Pb electronic structure hybridizes with the BP substrate. Pb thin films have been studied on metallic substrates, for example, grown on Sb(111) and Cu(111), where the Pb film strongly hybridizes with the supporting surface. In these examples, there is clear evidence of quantum well states, yet the strong hybridization leads to an expected quasi-particle poisoning that suppresses superconductivity[43,44]. In the case of Pb on BP, the presence of hybridization does not quench superconductivity, equating this case to a weaker regime of hybridization, akin to the concept of pseudodoping described for van der Waals monolayers on metallic substrates[45]. Based on the concept that weak hybridization can sculpt the BdG quasiparticles, we developed a hybrid two-band superconductor model which considers an isotropic superconducting band (Pb), which can hybridize with a highly anisotropic non-superconducting band (BP) (see methods 4). To this end, we allowed only pure Pb states to form Cooper pairs, while the BP states can affect the Pb ones only on a single-particle level. In such a model, we can consider the $k$-dependent effects of hybridization to the Fermi surface and its impact on the BdG quasiparticle spectrum as depicted in Fig. 5a. The main effect related to the experimental data is the emergence of (i) an anisotropic gap $\Delta(\theta)$ which changes the pairing potential on certain parts of the Fermi surface, and (ii) an anisotropic density of states which can be described by a so-called weighting function, $w(\theta)$.

From this model, we derived an analytical expression for the superconducting gap which can account for all the relevant details seen in the experimental spectra. The relevant fitting procedure and parameters are discussed in the methods and Supplementary Table 1. In Fig. 5b, we show an experimental spectrum taken on a 7 ML Pb film (also shown in Fig. 2a) and compare it to fits using (a) a conventional isotropic BCS superconductor gap function (magenta), (b) an anisotropic gap with no weighting function (orange) and (c) an anisotropic gap together with a weighting function derived from the proposed hybrid two-band model (red). We find that the two-band model can account for the three measured effects described in Fig. 2, enabling quantification of

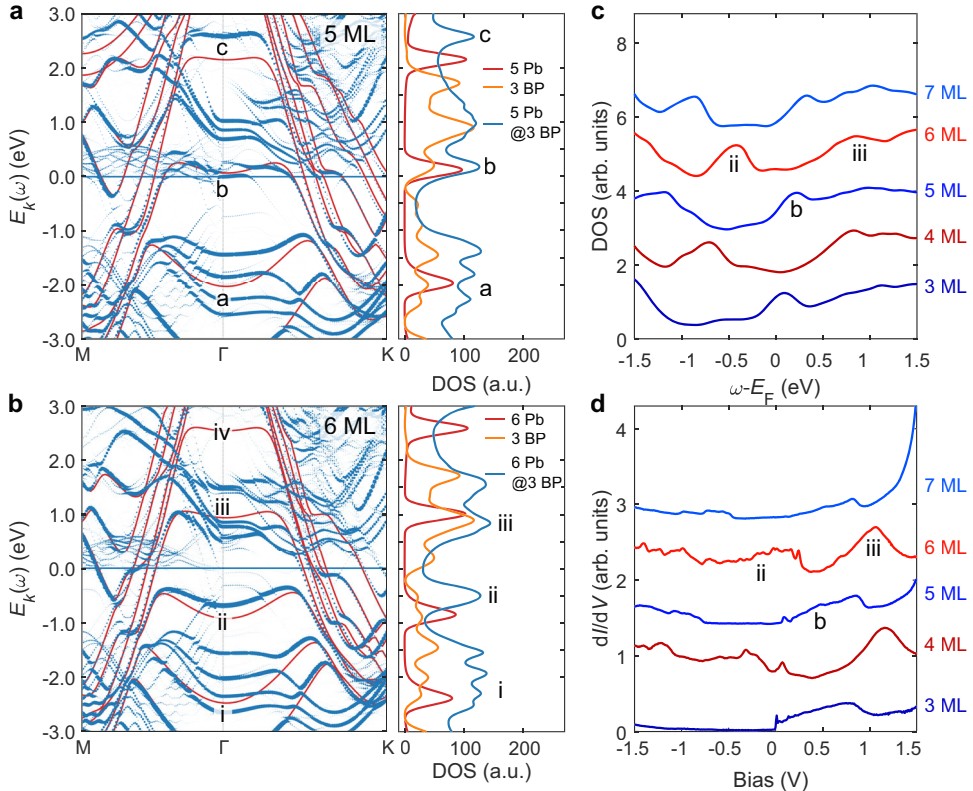

**Fig. 4 Electronic structure calculations of the hybrid structure and comparison to the experimental LDOS. a, b** Unfolded band structures for 5 ML (**a**) and 6 ML (**b**) Pb on 3 ML BP. These are plotted along high-symmetry points of the primitive Pb (111) unit cell. Solid red lines in the band structures refer to the pristine (but strained) 5 ML and 6 ML Pb band structure. Blue dots correspond to the unfolded spectral function as defined in Eq.(4) in the Supplementary Note 2, where the dot-size represents the intensity of $A(k,w)$ at the corresponding momentum and energy. The side panels of **a** and **b** show the quasi-local spectral functions $\rho(\omega)$ using a smearing $\sigma_k = 0.01$. Blue lines represent unfolded heterostructure data, while the red and orange lines depict $\rho(\omega)$ for the pristine Pb and BP structures, respectively. **c** Calculated spectra for Pb/BP heterostructures with various thicknesses of Pb films from 3 ML (bottom) to 7 ML (top). Films with an even (odd) number of MLs are color coded with red (blue) shades. Spectra are shifted vertically for clarity. **d** dI/dV spectra measured in wide bias range on Pb films with thickness from 3 ML (bottom) to 7 ML (top) grown on BP. Spectra are shown with color shades corresponding to the spectra in **c** ($V_{stab} = 1.5$ V, $I_{stab} = 500$ pA, $V_{mod} = 5$ mV, $T = 35$ mK).

both $\Delta(\theta)$ and $w(\theta)$, and their influence on the gap shape (Fig. 5c). By considering anisotropic effects alone, at low enough temperature, a step is expected to emerge in the spectral function which is not observed experimentally (see inset of Fig. 5b). Due to $w(\theta)$ the gap opening in the BdG quasiparticle dispersion is strongly suppressed in regions with enhanced hybridization as seen in the DFT calculations from Fig. 3b. Such weighting functions have been used in previous studies to phenomenologically describe observations of high temperature superconductors[46,47]. Here, we can directly derive the weighting behavior from the weak hybridization of states at the interface. Moreover, from our analytic model we can conclude that we need (a) an anisotropic Fermi surface yielding the weighting function $w(\theta)$ and (b) an anisotropic gap function $\Delta(\theta)$, which (c) must be in phase, i.e., $w(\theta)$ and $\Delta(\theta)$ are minimized/maximized for the same $\theta$, to be able to precisely reproduce the characteristics of the measured spectral function shown in Fig. 5b. While strain or anisotropic substrate screening could introduce anisotropic $w(\theta)$ and $\Delta(\theta)$ as well, these effects must not automatically yield $w(\theta)$ and $\Delta(\theta)$ to be in phase. Our suggested hybridization model accounts for this important property automatically.

We fitted all experimental data measured at $T = 30$ mK (Fig. 5d) and $T = 1.3$ K (Supplementary Figs. 12 and 13) with this model. We extracted $\Delta(N)$ for each synthesized film and found that $\Delta(N)$ is nearly thickness independent and does not show a clear bilayer oscillation. We attribute the latter to a

hybridization induced superconducting "deactivation" of the Pb QWS around $E_F$, such that only those parts of the Pb Fermi surface away from $\Gamma$ contribute to SC, which do not drastically change with the number of layers. Furthermore, we found similar values for the fitting parameters for both temperatures (Supplementary Table S1). This is not the case for anisotropic fitting models that neglect the weighting function. While the values of $\Delta$ do not change significantly for a given film, independent of thickness, we observed a significant variation in $\Delta$ from sample to sample. We suggest that these variations result from a gating effect stemming from a variation in the charge layer formed at the interface. Evidence of an accumulation layer can be seen in Supplementary Fig. 1, where spectroscopy at low voltages on individual islands shows a hard gap preventing tunneling through the bulk. Likewise, spectra measured on BP in the Pb vacancy islands show that the valence band edge is shifted in comparison to clean BP (Supplementary Fig. 1d), indicating an overall $n$-doping of the BP surface due to the Pb film. We find that cleaved BP is strongly $p$-doped and each cleave shows significant variations in the $p$-doping, as evidenced in spectroscopy of the band edges measured on the bare BP. From these measurements, the Fermi energy resides about 4–45 mV near the valence band edge (see Supplementary Fig. 14 and ref. [22]). Such variations in the Fermi energy will significantly affect the accumulation layer formed when Pb films are grown, leading to the suggested gating effect.

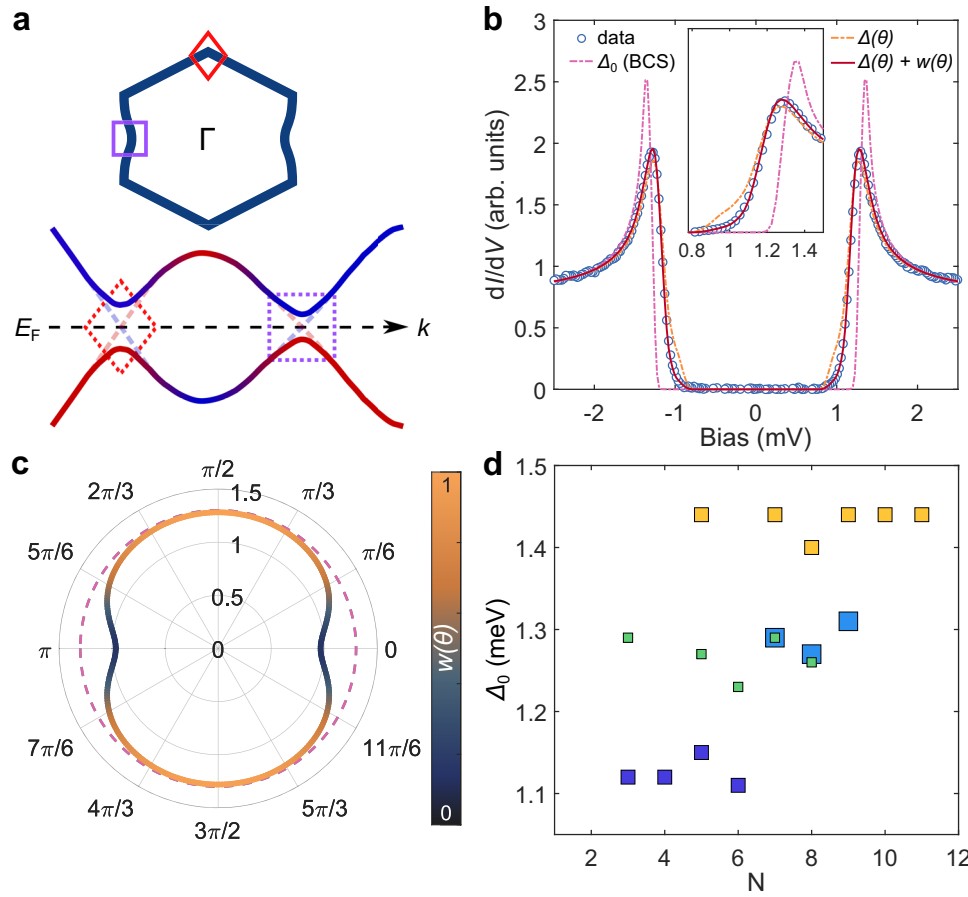

**Fig. 5 Modeling the hybrid superconducting gap and extracting the anisotropic weighting function. a** Top: sketch of the distorted Fermi surface for Pb/BP showing the K (diamond) and M (square) points. Bottom: schematic band structure for the hybrid two-band model showing the BdG band dispersion and opening of an anisotropic gap. Dashed lines represent hybridized bands in the normal state. Dashed diamond and square shapes in the diagram represent the K and M points of the Fermi surface, respectively. **b** d$I$/d$V$ spectrum (open blue circles) measured at $T = 30$ mK on a 7 ML film (same as in Fig. 2a) together with the fit employing a two-band model (red curve). The fit uses an anisotropic gap ($\Delta(\theta)$) and an anisotropic density of states represented by a weight function ($w(\theta)$) as shown in the polar plot in **c**. The magenta and orange curves in **b** represent simulated spectra with an isotropic gap (dashed line in **c**) and an anisotropic gap without incorporating the weight function, respectively. **d** $\Delta_0(N)$ obtained from fitting plotted for different sample preparations measured at $T = 30$ mK. The different colors of scatter points represent different preparations.

## Discussion

In conclusion, we demonstrated the creation of hybrid superconductivity driven by the interfacial coupling between Pb quantum films and BP. The signature of hybrid superconductivity is manifested by the anisotropic renormalization of the superconducting gap and its vortex structure. The anisotropic character of the superconducting properties can be traced to the anisotropic electronic structure of BP, which imprints itself on the hybrid Fermi surface and delocalizes the confined Pb wavefunctions. This hybridization results in a weighted and anistotropic distribution of the superconducting quasiparticles, whereby the order parameter develops a $k$-dependent amplitude. We quantify this weighting through an analytical hybrid two-band model, which enables direct and quantitative fitting to the anisotropic gap and allows for the extraction of the angle-dependent weighting function. It remains to be understood how the detailed moiré structure is responsible for the resultant superconducting behavior. For example, further studies are needed to better characterize the anisotropic vortex structure, and its dependencies on the structural and electronic properties of the grown films. We observe that near 30 ML thickness of the Pb film this behavior is suppressed, distinguishing a limit where we have interface-driven superconductivity, from a limit where

the interface is a weaker perturbation to the band structure of lead. The behavior observed and explained here demonstrates that the interfacial electronic structure provides a new degree of freedom to manipulate the superconductivity in superconductor-semiconductor heterostructures, which goes beyond considering the coupling of bulk bands between dissimilar materials. Based on this, we suggest that back gating the anisotropic substrate, i.e. here BP[48] or other anisotropic van der Waals semiconductors like ReSe$_2$[49], is a promising future step. As the renormalization of the QWS and the Fermi surface is sensitive to the relative band offsets between the metal and semiconductor, back gating can provide an efficient tuning parameter to control the intricate role of the hybridization on the resultant superconductivity and confined wavefunctions. These conclusions are important for the development of quantum technologies, which rely on lower dimensional superconducting nanostructures, illustrating that the interface electronic structure can play a critical role and ultimately be used to design the superconducting properties.

## Methods

**Experimental techniques.** All the scanning tunneling microscopy/spectroscopy (STM/S) measurements were performed in two home-built systems with base

temperatures of 30 mK[50] and 1.3 K[51]. Both systems are equipped with UHV chambers for sample preparation as well as a cold stage (~100 K) for the two-stage growth. We used black phosphorus (BP) crystals substrate which cleave easily along [001] direction as the van der Waals interaction between P layers is weak. Imaging such cleaved surfaces showed atomically flat topography within the scan range. Prior to the Pb film growth, black phosphorous (BP) crystals were first cleaved in-situ and transferred to the cold stage for the low temperature deposition. Pb was then evaporated from a Knudsen cell with a constant rate of 0.2 ML per minute. After the desired thickness was achieved, the sample was annealed shortly (~3–5 min) by resting the sample on the transfer arm at room temperature and subsequently inserted into the STM head for the low temperature measurements. Electronic properties of the samples were then measured using an electrochemically etched W or Cr tip which is coated with Au by dipping into a Au(111) crystal to minimize any spin polarization. For the scanning tunneling spectroscopy measurements, a standard lock-in technique was employed where a typical modulation voltage of $V_{\mathrm{mod\,(rms)}} = 20{-}100\,\mu V$ ($f = 877$ or 893.7 Hz) was added to bias and applied to the sample. To facilitate tunneling at low temperature, we made side contacts on the BP crystals using conducting epoxy. Supplementary Fig. 2 shows a typical low temperature growth showing closed Pb film and thicknesses from 7 ML to 11 ML. At a few locations of the sample, the film shows holes all the way to the BP crystal. Error bars are described where relevant.

**Ab initio calculations**. All ab initio calculations were performed using the Vienna Ab Initio Simulation Package (VASP)[52,53] utilizing the PBE exchange functional[54] within Projector Augmented Wave basis sets (PAW)[55,56]. The energy cutoff was set to 318.8 eV and we used for all (supercell) calculations $12 \times 12 \times 1$ ($3 \times 12 \times 1$) k grids.

**Supercell geometry**. To simulate heterostructures consisting of several monolayers of Pb on the BP substrate we construct minimal lateral supercells with six primitive Pb unit cells per four BP unit cells. This corresponds to three rectangular Pb unit cells (spanned by $a_x \times a_y$) on top of four BP unit cells (spanned $b_x \times b_y$). A top view of such a structure consisting of one monolayer Pb on a BP monolayer is depicted in Supplementary Fig. 8a. For the construction of these cells, we first optimized the lateral lattice constant of a five-layer Pb film yielding $a_0 \approx 3.57$ Å. Afterwards we applied the experimentally verified 1% strain in $y$-direction ([110] direction of the primitive Pb triangular lattice) yielding $a_x \approx 6.204$ Å and $a_y \approx 3.534$ Å. To model the bulk BP substrate, we use three monolayer of BP separated by $b_z = 5.3$ Å and set its lattice constant in $y$-direction to $b_y = a_y$ to match the minimal rectangular Pb unit cell in this direction, as depicted in Supplementary Fig. 8b. The lattice constant of BP in $x$-direction was set to $4b_x = 3a_x$ to create the minimal moiré cell. This way we get numerically feasible supercells with 78 (90) atoms for five (seven) monolayers of Pb while applying a strain of about 6.6% and 1.1% in $y$ and $x$ direction to the BP substrate[57]. The Pb atomic positions on the fixed BP substrate are finally relaxed until all forces acting on Pb atoms were <2.5 meV/ Å. This yields an average Pb-BP separation in z-direction of $d \approx 3.3$ Å. To minimize spurious interactions in $z$-direction due to the applied periodic boundary conditions we use a supercell height of $h \approx 45$ Å.

**Hybrid two-band model**. To describe the combined heterostructure of the Pb film and the BP substrate, we utilize a minimal two-band model of the form

$$H = \sum_{k,\sigma} \xi_k c_{k\sigma}^\dagger c_{k\sigma} + \sum_{k,k'} V_{kk'} c_{k\uparrow}^\dagger c_{-k\downarrow}^\dagger c_{-k'\downarrow} c_{k'\uparrow} + t\sum_{k,\sigma}\left[c_{k\sigma}^\dagger b_{k\sigma} + \mathrm{h.c.}\right] + \sum_{k,\sigma}\eta_k b_{k\sigma}^\dagger b_{k\sigma}.$$
(1)

Here, $c_{k\sigma}^\dagger\,(c_{k\sigma})$ and $b_{k\sigma}^\dagger\,(b_{k\sigma})$ are Pb and BP electron creation (annihilation) operators, $\xi_k$ and $\eta_k$ represent the isotropic and anisotropic electronic dispersions of Pb and BP, respectively, and $t$ describes a finite local hybridization between the distinct states. Most importantly, we allow only within the Pb states for a finite BCS-like Cooper pairing parameterized by $V_{kk'}$. Upon mean-field decoupling of the latter term and by defining the SC order parameter, i.e. the gap function $\Delta_k = \sum_{k'} V_{kk'}\langle c_{-k'\downarrow}c_{k'\uparrow}\rangle$, we get

$$H_{\mathrm{MF}} = \sum_{k,\sigma}\xi_k c_{k\sigma}^\dagger c_{k\sigma} + \sum_k\left[\Delta_k c_{k\uparrow}^\dagger c_{-k\downarrow}^\dagger + \mathrm{h.c.}\right] + t\sum_{k,\sigma}\left[c_{k\sigma}^\dagger b_{k\sigma} + \mathrm{h.c.}\right] + \sum_{k,\sigma}\eta_k b_{k\sigma}^\dagger b_{k\sigma}.$$
(2)

Within the extended Nambu-Gorkov space spanned by the spinor $\varphi_k^\dagger = \left(c_{k\uparrow}^\dagger, c_{-k\downarrow}, b_{k\uparrow}^\dagger, b_{-k\downarrow}\right)$ we can define the Green's function $G$

$$G^{-1} = \left[i\omega_n \mathbb{1} - H_{\mathrm{MF}}\right] = \begin{bmatrix} i\omega_n - \xi_k & -\Delta_k & -t & 0 \\ -\bar{\Delta}_k & i\omega_n + \xi_k & 0 & t \\ -t & 0 & i\omega_n - \eta_k & 0 \\ 0 & t & 0 & i\omega_n + \eta_k \end{bmatrix}.$$
(3)

By inverting $G$ and using the Nambu-Gorkov Dyson equation, we can analytically derive the explicit self-consistent gap equation reading

$$\Delta_k = -\sum_{k'} V_{kk'} \frac{\Delta_{k'}}{\sqrt{\left(E_{k'}^2 - \eta_{k'}^2\right)^2 + 4t^2\left[\Delta_{k'}^2 + (\eta_{k'} + \xi_{k'})^2\right]}}$$

$$\left[\frac{\eta_{k'}^2 - \lambda_-^2}{2\lambda_-}\tanh\left(\frac{\beta\lambda_-}{2}\right) - \frac{\eta_{k'}^2 - \lambda_+^2}{2\lambda_+}\tanh\left(\frac{\beta\lambda_+}{2}\right)\right],$$
(4)

with $E_k = \sqrt{\xi_k^2 + \Delta_k^2}$ and the full BdG quasiparticle dispersions

$$\lambda_\pm^2(k) = \frac{1}{2}\left[2t^2 + E_k^2 + \eta_k^2 \pm \sqrt{\left(E_k^2 - \eta_k^2\right)^2 + 4t^2\left[\Delta_k^2 + (\eta_k + \xi_k)^2\right]}\right].$$
(5)

In the limit of vanishing hybridization, i.e. $t = 0$, this reduces to the common BCS gap equation, as the two systems, Pb and BP, are fully disentangled. For finite $t$ however, we observe that both the gap equation and the BdG quasiparticle dispersions are affected. Thus, although only Pb states experience a finite pairing potential the resulting superconducting state is decisively affected by the normal but anisotropic BP states.

We proceed with analyzing the new BdG quasiparticle dispersions. To this end, we first introduce the normal dispersion of the hybridized band structure as resulting from the diagonalization of the single-particle terms of our two-band model

$$h_\pm(k) = \frac{\xi_k + \eta_k}{2} \pm \frac{1}{2}\sqrt{(\xi_k - \eta_k)^2 + 4t^2}.$$
(6)

With Eq. (6) we can expand $\lambda(k,\theta)$ in polar coordinates and in the weak hybridization limit as

$$\lambda_-^2(k,\theta) \simeq h_-^2(k,\theta) + \Delta^2(k,\theta),$$
(7)

where

$$\Delta^2(k,\theta) = \Delta_0^2\left[1 - \frac{\eta(k,\theta)t^2}{[\eta(k,\theta) - \xi(k,\theta)]^2[\eta(k,\theta) + \xi(k,\theta)]}\right].$$
(8)

The expansion parameter is here $\epsilon_k^2 = \frac{4t^2[\Delta^2 + (\eta + \xi)^2]}{(E^2 - \eta^2)^2}$, and we only require it to be small when evaluated at the original Fermi surface, i.e. $\xi = 0$. This condition is therefore satisfied when the energy gap $\Delta$ and the hybridization $t$ are small compared to the absolute value $\eta(k_F)$. This is fulfilled if two bands do not cross at the Fermi surface of Pb.

We evaluate $\Delta$ at the hybridized Fermi surface $k_F'(\theta)$, defined by the condition $h_-(k_F'(\theta)) = 0$. This yields

$$\Delta(\theta) \simeq \Delta_0\left[1 - \frac{t^2}{\eta^2(k_F'(\theta))}\right] = \Delta_0\left[1 - \frac{\xi^2(k_F'(\theta))}{t^2}\right].$$
(9)

The approximation results from expansion in the weak hybridization limit, identifying the small parameter $\frac{t}{\eta} \ll 1$, and neglecting higher orders in $t$. Note that in the $t = 0$ limit we correctly recover the effective gap $\Delta = \Delta_0$. From this we see that even in the presence of a constant pairing amplitude $\Delta_k = \Delta$, the hybridization at the interface can lead to an anisotropic effective energy gap $\Delta(\theta)$. Furthermore, the effective gap $\Delta(\theta)$ is always smaller than or equal to the initial gap $\Delta_0$. The minima of the effective gap are in fact to be found where $\xi(k_F')$ deviates the most from zero, i.e. where the hybridized Fermi surface deviates from the original Fermi surface of the single band in the Pb thin film. In these regions the superconducting states from the Pb band are mostly mixed with *non-superconducting* states from the band in the semiconductor substrate, thereby reducing the energy gap.

With this, we can map the two-band SC model to a single anisotropic SC band with effective anisotropic energy gap. The hybridization between the two bands is then ultimately responsible for both the imprinting of the anisotropic character to the final single band model—therefore determining a weighting function $w(\theta)$ [See Supplemental Note S3]—as well as for the anisotropy in the effective gap $\Delta(\theta)$. How the single anisotropic band and the effective gap are related to the two bands is to be determined by the choice of a specific model.

**Deducing weighting function and fitting parameters from the hybrid two-band model**. To describe our two-band system, we used an isotropic quadratic dispersion $\xi_k$ for the Pb band and an anisotropic quadratic dispersion $\eta_k$ for the BP band. In polar coordinates we have

$$\xi(k,\theta) = \frac{1}{2m^*}k^2 - \mu \quad \text{and} \quad \eta(k,\theta) = \frac{1}{2m_x^*}k^2 F(\theta) + \delta - \mu,$$
(10)

where $m^*$ describes the isotropic effective mass of Pb, $F(\theta) = 1 + \epsilon \sin^2(\theta)$ controls the anisotropy in the BP band $\epsilon = \frac{m_x^*}{m_y^*} - 1$, and $\delta$ represents the energy offset between the Pb and BP bands. For this model, the Fermi wavevector in the

hybridized system and in the weak hybridization limit reads

$$k_F'(\theta) = \left[2m^*\mu\left(1 + \frac{t^2}{\mu^2}\frac{1}{\frac{\delta}{\mu} - 1 + \frac{m^*}{m_x^*}F(\theta)}\right)\right]^{\frac{1}{2}} = k_0\left(1 + \frac{p^2}{\sigma - 1 + \chi F(\theta)}\right)^{\frac{1}{2}}, \quad (11)$$

where we introduced the parameters:

$$p = \frac{t}{\mu}, \; \sigma = \frac{\delta}{\mu}, \; \chi = \frac{m^*}{m_x^*}. \quad (12)$$

Here $k_0$ denotes the isotropic Fermi wavevector in Pb before hybridization. With this we can identify the weighting function in the hybridization model

$$w(\theta) = \left(\frac{k_F'(\theta)}{k_0}\right)^2 = 1 + \frac{p^2}{\sigma - 1 + \chi F(\theta)}. \quad (13)$$

With Eq. (9) and the Fermi vector from Eq.(11) we obtain:

$$\triangle(\theta) = \Delta_0\left[1 - \frac{p^2}{\left[\sigma - 1 + \chi F(\theta)\right]^2}\right]. \quad (14)$$

**Modeling STS spectra with hybrid two-band model.** In order to determine the superconducting gap structure, we numerically fit experimental d$I$/d$V$ curves to differential tunneling conductance derived as below.

The tunneling current between the tip and sample is given by,

$$I(U, T) \propto \int_{-\infty}^{\infty} \rho_t(E)\rho_s(E + eU)\left[f(E + eU, T) - f(E, T)\right]dE, \quad (15)$$

where, $\rho_s(E)$ is the sample density of states, $\rho_t(E)$ is the tip density of states and $f$ is the Fermi function. We used the Maki formalism to obtain the sample density of states[32,58] and modified it to include the anisotropic superconducting gap ($\Delta(\theta)$) and an anisotropic weighting function ($w(\theta)$) given by Eqs. (13) and (14),

$$\rho(\theta, \omega) = w(\theta)\cdot\Re\left(\frac{u}{\sqrt{u^2 - 1}}\right)$$

$$\text{with}, u = \frac{\omega}{\Delta(\theta)} + \zeta\frac{u}{\sqrt{1 - u^2}}. \quad (16)$$

Here, $\zeta$ is the pair breaking term which accounts for the weak suppression of the coherence peaks in our system. The resulting differential conductance is given as ref. [59],

$$\frac{dI}{dU}(U, T) \propto \int_{-\pi}^{+\pi}\int_{-\pi}^{+\pi}\sin(\alpha)I\left(U + \sqrt{2}V_{mod}\sin(\alpha), \theta, T\right)d\alpha d\theta \quad (17)$$

where, $V_{mod}$ is the bias modulation used for the STS measurements. Here, the additional integration over $\alpha$ accounts for the broadening in d$I$/d$V$ caused by the modulation voltage. Supplementary Fig. 12 shows fits to d$I$/d$V$ spectra measured on different samples, with extracted parameters summarized in Supplementary Table 1. The standard error in each parameter is <5%.

## Data availability

All the data supporting results are available within the paper and supplementary information or from the corresponding authors upon request.

## Code availability

Codes used for the data analysis are available on request from the corresponding authors.

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

## Acknowledgements

The experimental part of this project was supported by the European Research Council (ERC) under the European Union's Horizon 2020 research and innovation programme (grant no. 818399; SPINAPSE). We also acknowledge support from the NWO-VIDI project 'Manipulating the interplay between superconductivity and chiral magnetism at the single-atom level' with project no. 680-47-534. M.I.K. acknowledges support by European Research Council via Synergy Grant 854843—FASTCORR. We also acknowledge funding from Microsoft Quantum. M.R. acknowledges helpful discussions with Alexander Rudenko. M.R. thanks SURF (www.surf.nl) for the support in using the National Supercomputer Snellius.

## Author contributions

A.K., E.S, M.St., U.K., and E.J.K. performed the experiments. M.R and A.A.K. designed the experiments. M.Si., M.I.K., and M.R. performed the ab initio calculations and developed the hybrid two-band model. A.K., E.S., E.J.K., P.K and A.A.K. performed the experimental analysis. All authors participated in the scientific discussion of the results, as well as participated in writing the manuscript.

## Competing interests

The authors declare no competing interests.
