## [Peer Review File · Nature Communications]

Tuning lower dimensional superconductivity with hybridization at a superconducting-semiconducting interfaceREVIEWER COMMENTS

Reviewer #1 (Remarks to the Author):

The authors report a STM study of the superconductivity in a Pb-black phosphorus (BP) hybrid structure. They found the superconductivity is strongly anisotropic. The main observations are the renormalized superconducting gap and the vortex structures. The authors attribute the origin of the anisotropy to the electronic structure of BP, and an analytical model has been used to understand the experimental spectra.

I find the manuscript interesting, and it provides a new superconducting-semiconducting interface, in which the superconductivity can be tuned to some extent. However, since the observed renormalization of superconductivity is tiny, I don't think these results can make significant impacts on the community. I therefore cannot recommend the publication of this work in Nature Communications. I would rather see it in a more specialized journal.

1. I don't understand the statement, the gaps "can be characterized by a strongly broadened and V-shaped gap structure, beyond a full superconducting gap (In page 4)". In Fig. 2A, all the spectra are U-shaped.
2. In Fig. 2C and D, the selected two directions are arbitrary.
3. The spectra in Fig. 3C are rather noisy. Given the high-quality superconducting gaps in Fig. 2A, it is hard to understand why the quality of the spectra in Fig. 3C and Fig. S8 is not good.
4. In Page 8, the authors claimed "We observed a clear variation in Δ from sample to sample". Such variation is due to the quality of the sample or other physics behind need to be reconsidered?

Reviewer #2 (Remarks to the Author):

Key Results

The study of structural and electronic phenomena occurring at the interfaces between two contacting materials are of the most attractive areas of research in surface science. Growing thin superconducting thin films on the surface of semiconductors, semi-metals, or topological insulators are of great interest, since the interface driven effects can induce peculiar properties in superconducting material due to the proximity.

The present work studies the properties of a 2D superconducting Pb thin films grown on a strongly anisotropic narrow-band semiconductor BP. The authors observed the following interface-induced effects:

1. The moiré pattern in the Pb film up to 30 ML thickness, caused by lattice mismatch at the interface
2. Nonthermal broadening of the the superconducting gap and quantum well states (QWS) for thin Pb films (≤ 12 ML) and no such broadening for Pb films > 30 ML
3. Anisotropic behavior of Caroli-de-Gennes Matricon states trapped in the vortex core in Pb/BP film, which are not characteristic of isotropic superconducting Pb.

The authors explain the observed results by the presence of weak coupling between of Pb and BP anisotropic bands at their interface, which modifies the Fermi surface of Pb and leads in turn to the anisotropy of the order parameter. The authors provide a model that describes the induced anisotropy and approximates well the observed nonthermal broadening of SC gap, that deviates from pure BCS gap.

Significance

Pb is a widely used superconductor for growing heterostructures because it is superconductive in very thin films down to several atomic layers. Therefore, the interface-induced changes in properties of the

superconducting Pb films, as well as proximity induced changes in substrate properties have been studied in various weakly coupled heterostructures:

1. Non-thermal distortion of the SC gap induced by weak coupled interface in Pb films grown on narrow gap semiconductor InAs [F. Magnus et al., Appl. Phys. Lett. 92, 012501 (2008)]
2. Undisturbed properties of Pb films with sharp QWS features on graphene [Phys. Rev. B 98, 195441 (2018)]
3. Pb film on topological insulator Bi₂Te₃, proximity modified properties of the surface states of the substrate [J. Phys. Chem. Lett. 12, 9068, (2021)]

also in strongly coupled:

4. Pb films on semimetallic Sb(111), where superconductivity is strongly modified by the underlying substrate [T. Vincent et al., Phys. Rev. B 98, 155440 (2018)]

The novelty of the authors' work lies not only in the choice of a new substrate, but also in the first proposed explanation of the anisotropy mechanism. Nevertheless, similar effects have been previously observed in experimental works on other substrates.

Validity

The authors based their model on the observed anisotropy of the Caroli-de-Gennes Matricon states trapped in the vortex core in Pb/BP film, which can be caused in weak magnetic fields by anisotropy of the pairing potential or anisotropy of the DOS at Fermi level [N. Hayashi et al., PRL 77, 4074 (1996)]. The introduced model, which resembles the proximity induced anisotropic Fermi surface, results in angle dependent order parameter and describes well the measured broaden SC gap in Pb thin films (with correctly chosen fitting parameters). To strengthen the fidelity of the proposed model, I could recommend the authors to simulate the LDOS of quasiparticles in the vortex core with similar fitting parameters and confirm the correspondence with the observed anisotropy of CdGM states in the experiment.

Data and methodology

The quality of the experimental results performed in the manuscript are at a high level, the data are clearly presented and analyzed. Some of the improvements in figures and explanations listed below could greatly improve following the authors' ideas in the text

Suggested improvements

It would be very helpful to link the K and M points of Pb Brillouin zone in Fig. 3 a& b with symmetry axis on the dI/dU map of the vortices in Fig. 2 (c).

It's not clear from the text nor from the image in Fig.2(c) the choice of lines along which the dI/dU spectra are measured and presented in Fig. 2 (d). It would be very helpful to present the dI/dU map of the vortex at bias voltages less than Δ/e (even close to zero bias), where the anisotropy should be seen more clearly.

The distortion of the Fermi surface near Γ point in Fig3. (b) is not discussed in the manuscript, however, to my knowledge, STM technique is mostly sensitive to tunneling events with k_{\parallel} close to Γ point. How much does it affect the end result?

I could not completely figure out what additional features in LDOS the authors are talking about in the the stroke #190 and what peaks they attributed to QWS in Fig.3 c & d. Figure S8 clearly shows a peak at ~ 1.2 V for the 4 ML film, which shifts slightly to the left as the film thickness increases with an

odd number of ML, which is consistent with QWS. The similar behavior I can guess for the 7ML, where the peak is beyond the energy range and is shifting to the left with increasing the thickness of the film. Do the authors have a wider energy range spectra for 3ML, 5ML and 7ML where this peak can be identified? Do these peaks have the same FWHH for the odd and even ML Pb films? Is it possible to add data for 8-9 ML films to the Fig. 3 c&d and mark the peaks corresponding with QWS.

Could the broadening of QWS might be caused by the in homogeneity of the film, which has varying thickness and stronger relative variation for thin films than for films >30ML?

Reviewer #3 (Remarks to the Author):

In this manuscript the authors investigate the superconducting properties of thin Pb films grown on a highly anisotropic black phosphorus (BP) substrate. It is found that the superconducting gap and the vortex structure of the thin film undergo an anisotropic renormalization as a result of the coupling to the BP substrate. The presence of a strong moiré lattice and a uniaxial strain of the Pb film are also reported. While the results of this work are potentially interesting, I find the presentation rather confusing and I believe that the manuscript should not be published in this form. A few specific points are detailed below.

The experimental observations reveal certain specific properties of this hybrid system and I believe that a key task is to understand the mechanism(s) responsible for the observed features. For example, the presence of the interface may induce electrostatic effects (band bending), strain (which, in turn, may affect the electronic properties), hybridization effects (which involves mixing of the electronic wave functions associated with the two subsystems), and various spin-dependent effects. The picture that emerges from the manuscript is rather murky. For example, it is suggested that, as a result of hybridization at the interface, the wave function in thin Pb films becomes delocalized. I assume that this implies that the electrons penetrate into the BP. If this is the case, how does one explain the absence of a proximity induced gap in the BP (at least near the interface)? What are the specific roles of electrostatic effects, strain, and hybridization? Note that the title suggests that the superconducting properties can be “tuned” using the hybridization at the interface. In this context, one should explicitly check that the observed properties cannot be generated by, e.g., a strained Pb lattice (i.e., without invoking the electronic properties of BP).

In addition, I think that some of the language should be more transparent (and exact). For example, what does it mean that the “hybrid electronic structure derived at the interface ... can be activated to tune the superconductivity in the lead film.”? In general, what is the meaning of “tuning” in this context? Also, what should one understand by the concept of “interface electronic structure”. Note that the experimental results concern the electronic properties of a thin Pb film, which are perturbed by the PB substrate. I do not exactly see in what sense one can view the “interfacial electronic structure” as “a new degree of freedom”. I would be much more comfortable to simply say that the effects of Pb-BP hybridization depend on the (Pb) film thickness.

The claim that the measured dI/dV features in Fig. 3B “compare well to the calculated DOS (Fig. 3D)” requires further explanation (since the correspondence is not obvious).

We would like to thank all referees for their constructive comments. Below, we answer point by point, to all specific replies. Where relevant, we highlight, where we have made changes to the manuscript.

Reviewer #1 (Remarks to the Author):

The authors report a STM study of the superconductivity in a Pb-black phosphorus (BP) hybrid structure. They found the superconductivity is strongly anisotropic. The main observations are the renormalized superconducting gap and the vortex structures. The authors attribute the origin of the anisotropy to the electronic structure of BP, and an analytical model has been used to understand the experimental spectra.

I find the manuscript interesting, and it provides a new superconducting-semiconducting interface, in which the superconductivity can be tuned to some extent. However, since the observed renormalization of superconductivity is tiny, I don't think these results can make significant impacts on the community. I therefore cannot recommend the publication of this work in Nature Communications. I would rather see it in a more specialized journal.

We thank the referee for the expression of novelty. However, we disagree that the renormalization of superconductivity is tiny as tiny would imply that the renormalization is negligible. The renormalization of superconductivity we observed is clearly non-negligible:

(i) The superconducting energy gap is strongly renormalized and varies between 1.1 – 1.6 meV, depending on the film growth. This amounts to about 23 % change with respect to the bulk value, which is clearly not negligible.

(ii) The superconducting gap shape is renormalized, as we detail in the paper. This effect cannot be ascribed to a thermal broadening effect, as this apparent broadening has three distinct effects: a) it rounds the gap onset, b) it changes the coherence peak height, and c) it broadens the step (i.e. what we refer to as V-shaping). These deviations are plotted in Figs. 2A and 4B (now Fig. 5B), where it is directly compared to experimental data for Pb on Si (Fig. 2A), as well as fitted with various models which exclude the V-shaping [Fig. 4B (now Fig. 5B)].

(iii) We observed strongly anisotropic vortices, which deviate decisively from the expected isotropic vortices seen for typical thin Pb films. These vortices resemble those in materials like NbSe₂ rather than typical Pb films (Fig. 2) and cannot be accounted for if the renormalization of the superconductivity is considered negligible.

In addition to this, the QWS and the quantum size effects on growth are strongly modified as we described in the report. We believe this evidence strongly refutes the statement that this is tiny/negligible.

1. I don't understand the statement, the gaps "can be characterized by a strongly broadened and V-shaped gap structure, beyond a full superconducting gap (In page 4)". In Fig. 2A, all the spectra are U-shaped.

We strongly disagree with the last statement and the deviations from a U-shaped gap are detailed in the manuscript. However, the wording may have been incorrect in our initial submission and we believe could have led to confusion. To be precise, we do not claim that the gap structure is V-shaped, but that there is a non-thermally induced structural renormalization to the gap edge, leading to an apparent V-shaping of the gap structure on top of a full gap along with other effects mentioned above. **We have removed the word V-shaped from our manuscript.** Exhaustive care was taken to clearly illustrate this in the revised version of our manuscript:

At the outset of this work, we spent exhaustive time trying to fit the experimental data considering that the gap can be described by a typical gap derived from standard BCS fitting models (e.g. Dynes broadening or Maki fitting). These models simply do not fit the data well and yield unrealistic values (e.g. oscillating values of effective temperature and lifetime broadening). To further elaborate: the

spectra for Pb on BP in the ultra-thin limit cannot be fitted with a U-shaped gap using BCS theory, including any traditional considerations. These different fits are illustrated in Fig. 4B (Fig.5B), where a comparison to an expected U-shaped curve (red dashed curves) is made to the renormalized gap. The renormalized gap structure based on our two-band model fits the experimental data whereas the U-shaped structure cannot recapture the three different changes seen in the typical gap structure mentioned above. This is the reason for the development of our two-band model, and the fitting procedure shown in Fig. 4 (Fig 5).

To further illustrate the gap renormalization, the spectra measured on Pb on BP was directly compared to Pb films grown on Si(111) which show a U shape. It is illustrated in Fig. 2A that there is a non-thermal broadening of the slope of the spectra (as well as changes to the onset and coherence peak intensities) that cannot be captured by conventional BCS fitting. These modulations which we refer to as the renormalization of the gap structure are not visible in (i) thicker films of Pb on BP, and (ii) on Pb/Si(111).

To make sure to avoid confusion between claiming a V-shaped gap, and a reshaping of the gap, we removed the text referencing a V-shaped gap.

2. In Fig. 2C and D, the selected two directions are arbitrary.

We do not follow the question or concern here. We chose these two directions for the given vortex such that the spectra are taken on a flat area with the same thickness, and to avoid the island edges. The selected directions are not parallel to each other, albeit not orthogonal. The line cuts show that the gap shape has a variation in magnetic field in the two different directions (see Fig. 2D). One direction was chosen to be aligned with a high symmetry axis of Pb (along [110] of Pb lattice). This illustrates the key point: the superconducting gap in magnetic field is not isotropic, and, as stating above, therefore the gap renormalization is not negligible.

In the revised Fig. 2, we have labeled the crystal orientations for better reference. Also, we added the direction for lines with respect to the Pb lattice and added following sentence on page 6

The dI/dV spectra measured along one of the extended stripes (along [110] of Pb lattice) shows a small splitting of the zero bias peak away from the center and extends over several 10's of nanometers (direction 2) before abruptly merging into coherence peaks. However, dI/dV spectra measured in a nearly perpendicular direction (direction 1) show a splitting of the zero bias peak and smooth merging into the coherence peaks above 10 nm. We chose these two directions for the given vortex such that the spectra are taken on a flat area with the same thickness, and to avoid the island edges.

3. The spectra in Fig. 3C are rather noisy. Given the high-quality superconducting gaps in Fig. 2A, it is hard to understand why the quality of the spectra in Fig. 3C and Fig. S8 is not good.

The spectra are not noisy, but are rather an accurate representation of how the spectra typically look for this material system. Our detection limit and noise level are far below the measured variations, and simply said, the spectra show more structure due to the hybridization compared to QWS of Pb on wide-gap materials like Si. Moreover, we have measured these spectra on two different experimental setups with extremely high resolution and low noise conditions, and many different films, as well as the same high-resolution setup that was used to measure the superconducting gaps.

In Fig. 3C (Fig. 4D), the finer variations in the measured spectra for varying energies are highly reproducible (we have studied nearly 40 different films). To substantiate this point, we plot below line spectroscopy data measured on 6 and 7ML films from different growths (B-C) as well as spectra measured on various thicknesses for the single growth (D-F). Spectra are shifted vertically for clarity and each color shade represents the same sample growth. For the first data set, it can be seen that the characteristic features on a large energy scale are seen on different samples, with different tips, with small variations in intensity. For a given film, the second data sets show that the spectral features are nearly independent of location, where again only small variations are seen in intensity. The spectroscopy data clearly illustrate that these features in the spectra are reproducible and not attributed to noise.

Fig. R1: Spectroscopy along a line measured on various Pb films and thicknesses. (A) Constant current STM image of the Pb film showing typical line profile for spectroscopy measurements. ($V_s = 600$ mV, $I_t = 10$ pA, scale bar = 10 nm). (B-C) dI/dV spectra averaged over a line on 6 ML and 7 ML Pb films for different growths. (D-F) dI/dV spectra measured along a line for the same sample but various thicknesses as indicated in the panel. STS parameters for panels B, C: $V_{\text{stab}} = 1.5$ V, $I_{\text{stab}} = 200$ pA, $V_{\text{mod}} = 20$ mV, $T = 1.3$ K. STS parameters for panels D-F: $V_{\text{stab}} = 1.5$ V, $I_{\text{stab}} = 200$ pA, $V_{\text{mod}} = 10$ mV, $T = 1.3$ K. All the spectra are shifted vertically for clarity and each color shade represents same sample growth.

Fig R2. Series of dI/dV spectra measured in a wide bias range for Pb islands (A) and Pb films (B) of thicknesses indicated on the left side of each curve. Even/odd films are plotted with same color shades (red/blue). Spectra are shifted vertically for clarity. STS parameters for panel A: $V_{\text{stab}} = 1.5$ V, $I_{\text{stab}} = 200$ pA, $V_{\text{mod}} = 10$ mV, $T = 1.3$ K. STS parameters for panel B: $V_{\text{stab}} = 1.5$ V, $I_{\text{stab}} = 200$ pA, $V_{\text{mod}} = 20$ mV, $T = 1.3$ K.

Spectroscopic features observed on thin films characteristics to different layer thicknesses are also persistent on Pb islands which are grown at room temperature (Fig. R2).

We have added these two data sets to the supplement (Fig. S8 and Fig. S9). We have also added following sentence to the manuscript on page 8:

We note that the finer variations seen in the spectroscopy are consistent between sample growths, for films of the same thickness, and do not significantly vary on a given terrace (i.e. do not show strong spatial variations) (Fig. S8). These features are also persistent on Pb islands grown at room temperature, enabling a clear spectroscopic fingerprinting of given Pb layers (Fig. S9).

4. In Page 8, the authors claimed “We observed a clear variation in Δ from sample to sample”. Such variation is due to the quality of the sample or other physics behind need to be reconsidered?

This is an interesting question, and we have a hypothesis for this variation, as we detail in the next paragraph. To start with the experimental observation: we see that there are relatively small and negligible variations in the gap spectra for a given sample, both in terms of spatial variation and varying thickness. However, we see a large variation in the value of the superconducting gap, from sample to sample.

Pb *n*-dopes the BP surface, which is natively *p*-doped. This leads to a charge layer and prevents us from tunneling through the bulk at low voltages. This is why we created closed films and contacted the surface of the material. We detail this in the revised manuscript on page 5 and in the newly added Fig. S13. In these figures, we show islands of Pb grown where bare BP is exposed. One can see that there is a hard gap at low voltage, which hampers the ability to tunnel through the bulk of the sample (Fig. 13B). Likewise, the spectra show that the valence band edges are shifted in comparison to clean BP, indicated an overall *n*-doping of the surface due to the presence of Pb. This observation indicates that there is an accumulation layer near the BP/Pb interface, which creates a weak insulating barrier.

We observe that the Fermi energy of BP varies from sample to sample. As the band bending at the interface between Pb and BP is highly sensitive to the Fermi energy of BP, this leads to a variation in the Pb-induced accumulation layer near the BP surface. We find that cleaved BP is strongly *p*-doped, with the Fermi energy residing about 4 - 45 mV near the valence band edge (see ref. Kiraly *et al Nano Letters*, (2017), and Fig. S14). This variation in the accumulation layer can lead to a small gating effect, and we hypothesize that such effects can modulate the superconductivity. This is an exciting finding, which we will further investigate by looking at variations in doping of the underlying BP in the future.

To summarize, we have improved the discussion of this effect on page 11 in the paragraph before conclusions and added Fig. S13 to detail these effects.

Reviewer #2 (Remarks to the Author):

Key Results

The study of structural and electronic phenomena occurring at the interfaces between two contacting materials are of the most attractive areas of research in surface science. Growing thin superconducting thin films on the surface of semiconductors, semi-metals, or topological insulators are of great interest, since the interface driven effects can induce peculiar properties in superconducting material due to the proximity.

The present work studies the properties of a 2D superconducting Pb thin films grown on a strongly anisotropic narrow-band semiconductor BP. The authors observed the following interface-induced effects:

- 1. The moiré pattern in the Pb film up to 30 ML thickness, caused by lattice mismatch at the interface*

2. Nonthermal broadening of the the superconducting gap and quantum well states (QWS) for thin Pb films ($\leq 12\text{ML}$) and no such broadening for Pb films $> 30\text{ML}$

3. Anisotropic behavior of Caroli-de-Gennes Matricon states trapped in the vortex core in Pb/BP film, which are not characteristic of isotropic superconducting Pb.

The authors explain the observed results by the presence of weak coupling between of Pb and BP anisotropic bands at their interface, which modifies the Fermi surface of Pb and leads in turn to the anisotropy of the order parameter. The authors provide a model that describes the induced anisotropy and approximates well the observed nonthermal broadening of SC gap, that deviates from pure BCS gap.

We thank the referee for this clear summary of the main points of our manuscript.

Significance

Pb is a widely used superconductor for growing heterostructures because it is superconductive in very thin films down to several atomic layers. Therefore, the interface-induced changes in properties of the superconducting Pb films, as well as proximity induced changes in substrate properties have been studied in various weakly coupled heterostructures:

1. Non-thermal distortion of the SC gap induced by weak coupled interface in Pb films grown on narrow gap semiconductor InAs [F. Magnus et al., Appl. Phys. Lett. 92, 012501 (2008)]

2. Undisturbed properties of Pb films with sharp QWS features on graphene [Phys. Rev. B 98, 195441 (2018)]

3. Pb film on topological insulator Bi₂Te₃, proximity modified properties of the surface states of the substrate [J. Phys. Chem. Lett. 12, 9068, (2021)]

also in strongly coupled:

4. Pb films on semimetallic Sb(111), where superconductivity is strongly modified by the underlying substrate [T. Vincent et al., Phys. Rev. B 98, 155440 (2018)]

The novelty of the authors' work lies not only in the choice of a new substrate, but also in the first proposed explanation of the anisotropy mechanism. Nevertheless, similar effects have been previously observed in experimental works on other substrates.

We thank the referee for these references. We have added them to the revised manuscript along with a discussion about their significance in juxtaposition with our findings 2. While these works are relevant for the discussion about thin-film superconductivity and thin-film Pb, they do not capture the main aspects of our manuscript either experimentally or theoretically. Also, there is no account in these papers of the theoretical modeling we illustrate.

Ref. 1: The devices made were point contact junctions designed for performing Andreev reflection spectroscopy and the interface between the superconductor and semiconductors are punctured by various means to control the barrier transmission. The interfaces are poorly defined and the films are relatively thick. Therefore, parameters extracted from spectroscopy on these devices represents characteristics of the junctions rather than the interface and its hybridization. Specifically, the authors suggest that the smearing seen in the devices are due to inhomogeneities, disorder and a thick interface. Therefore, they are not relevant to our paper. The only relevant devices are the ones made with route 3(i) with a high Z barrier. However, the authors only speculate its origin to inelastic scattering and they do not discuss the role of interface.

Ref. 2: We are aware of an earlier work, where the QWS of Pb are studied on graphene (also see J. H. Dil et al., PRB **73**, 161308(R) (2006)). In this work, Pb is claimed to be more free-standing compared to other semiconductors, which would be the limit of no coupling. We have added this citation to our paper, but from the QWS standpoint, this is a different limit than the one we study.

Ref. 3: This recent work focused on the proximity-induced superconductivity on the Bi_2Te_3 surface. In the case of the n -doping we observe for Pb on BP, we do not see any proximity-induced superconductivity in the BP (see SI S3 and reply to Referee 1). Likewise, we observe no wetting layer growth in our work (see SI Fig. S13 and Fig.S3). Finally, while the work is interesting, this manuscript does not focus on the changes of the superconductivity of the Pb itself and the role of the interface in modifying this.

Ref. 4: The superconductivity is quenched as it is grown on a metallic surface. Similar observations have been made early for Pb grown on Cu(111) (PRL 114, 047002 (2015)), and this would be an expected effect of quasiparticle poisoning. This is also an effect of hybridization, when hybridization is much stronger than what we investigate here. We agree that this hybridization can modify the QWS, and have accredited this in the manuscript. There is also another technical point: this work is performed on islands and not closed films. Therefore, the reason for vanishing of the superconducting gap may be either due to lateral quantum size effect or strong hybridization.

We have added a small discussion about all these pertinent references throughout the manuscript.

Specifically, we have added following sentences at corresponding places:

Ref. 1 on page 5: *Non-thermal broadening has been seen in point contact spectroscopy measurements in Pb/InAs tunnel junctions, however, such effect do not stem from the interface but rather from junction properties.*

Ref. 2 on page 3: *The hybrid electronic structure leads to fingerprints that can be seen from both calculations and STS measurements, that strongly differ from signatures of quantum well states (QWS) of nearly free-standing Pb films.*

Ref. 3 on page 5: *Pb leads to an n -doping of the BP near the interface, pushing the Fermi energy into the intrinsic bulk band gap and thus quenching any carriers in the BP that can be proximitized. This is in contrast to Pb grown on other narrow band gap semiconductors, such as Bi_2Te_3 , where the samples are intrinsically n -doped and exhibit a sizeable proximity effect.*

Ref. 4 on page 9: *Pb thin films have been studied on metallic substrates, for example, grown on Sb(111) and Cu(111), where the Pb film strongly hybridizes with the supporting surface. In these named examples, there is clear evidence of quantum well states, yet the strong hybridization leads to an expected quasiparticle poisoning that suppresses superconductivity.*

Validity

The authors based their model on the observed anisotropy of the Caroli-de-Gennes Matricon states trapped in the vortex core in Pb/BP film, which can be caused in weak magnetic fields by anisotropy of the pairing potential or anisotropy of the DOS at Fermi level [N. Hayashi et al., PRL 77, 4074 (1996)]. The introduced model, which resembles the proximity induced anisotropic Fermi surface, results in angle dependent order parameter and describes well the measured broaden SC gap in Pb thin films (with correctly chosen fitting parameters). To strengthen the fidelity of the proposed model, I could recommend the authors to simulate the LDOS of quasiparticles in the vortex core with similar fitting parameters and confirm the correspondence with the observed anisotropy of CdGM states in the experiment.

We thank the referee for this suggestion and agree that this would be an interesting theoretical extension of our study for the future. For the manuscript in its current / revised form, we however need to note that our hybridized two-band model is not based on the observed anisotropic CdGM states in the vortices, but rather on the layer dependent hybridization effects seen in the normal state STS data [Fig. 3 C (now Fig. 4D) and S8 (Fig. S8-10)], which we can accurately understand in comparison with our DFT calculations. From this experimental-theoretical data, we derive the relevant role of hybridization in the system which led to the introduction of the local hybridization term in the two-band model. This model (a) explains the physical origin of the weighting functions and (b) perfectly interpolates the measured STS data in the SC state with a minimal set of fitting parameters. We

therefore believe that we show enough evidence for the validity of the model and leave the suggestion by the referee, which is beyond the content of this manuscript, for future study.

Data and methodology

The quality of the experimental results performed in the manuscript are at a high level, the data are clearly presented and analyzed. Some of the improvements in figures and explanations listed below could greatly improve following the authors' ideas in the text.

Suggested improvements

It would be very helpful to link the K and M points of Pb Brillouin zone in Fig. 3 a& b with symmetry axis on the dI/dU map of the vortices in Fig. 2 (c).

We have added the crystal orientations now for reference in Fig. 2C.

It's not clear from the text nor from the image in Fig.2(c) the choice of lines along which the dI/dU spectra are measured and presented in Fig. 2 (d). It would be very helpful to present the dI/dU map of the vortex at bias voltages less than Δ/e (even close to zero bias), where the anisotropy should be seen more clearly.

The lines for dI/dV spectra were chosen in such a way that one of the line is parallel to a high symmetry axis of Pb(111) lattice and the other line is nearly perpendicular to it (see reply to reviewer 1 point #2). We chose these two directions for the given vortex such that the spectra are taken on a flat area with the same thickness, and to avoid the island edges.

Below, in the newly added Fig. S4, we show an example of a dI/dV map acquired near a coherence peak in the same area as Fig. 2B. At these imaging biases, we typically do not see the strong anisotropic character of the vortices. It is important to note that a number of STM-based studies of vortices illustrate that the anisotropic character of the vortex is only clearly visible by taking maps at zero bias. (see e.g. PRL, 62, 214 (1989), Phys. Rev. B 102, 174502 (2020), arXiv:2105.01354). It has been described in refs. (e.g. PRB 56, 9052 (1997), PRB 53, 15316 (1996)), that additional electronic structure can be seen near the gap edge making it difficult to disentangle the anisotropic character of the vortex. This is why we imaged the vortices at zero bias. We have added following requested map to the supplement (Fig. S4).

Fig. R3. Vortex imaging at different bias voltage. Simultaneously measured dI/dV map at $V = 0$ V (A) and $V = 1.3$ mV (B) at $T = 30$ mK and $H = 50$ mT, in the same area as Fig. 2B of the main manuscript. Imaging parameters $V_{\text{stab}} = 10$ mV, $I_{\text{stab}} = 10$ pA, $V_{\text{mod}} = 100$ μ V, $\Delta z = -80$ pm (scale bar = 200 nm)

We have added following sentence to the manuscript on page #6:

Simultaneous vortex imaging at coherence peak in the same area (Fig. S4) reveal circular shaped vortices, indicating a strong k -dependent dispersion of BdG quasiparticles.

The distortion of the Fermi surface near Γ point in Fig3. (b) is not discussed in the manuscript, however, to my knowledge, STM technique is mostly sensitive to tunneling events with $k//$ close to Γ point. How much does it affect the end result?

We thank the referee for raising this point. While our DFT calculations for the deformation of the Fermi surface away from Γ are most reliable, the details around Γ strongly depend on computational approximations. This is due to the peculiar characteristics of the quantum well states in Pb. As visible from the band structure calculations of pristine Pb films shown as red solid lines in Fig. S7 (now Fig. 4 A and B in the manuscript and as red dashed below) there are rather flat bands at Γ , which are close to E_F for odd layers and shifted away from E_F (and split) for even layers. These flat bands together with the tunneling sensitivity to states around Γ form the well-known QWS seen as well-defined peaks in STS with a bi-layer oscillation. The exact energetic positions of these QWS however strongly depend on the exact doping levels, the intrinsic strain, as well as on the chosen approximations within the DFT calculations. For the case of the 5 ML Pb, as the depicted example in Fig. 3A, the QWS around Γ is very close to E_F which yields together with the broadening a round feature in the Fermi surface map of Fig. 3A. As this QWS state strongly hybridizes with the BP conduction states in our calculations and as the depicted Fermi surfaces are taking all states in an energy windows of ± 33 meV around the Fermi level into account, the Fermi surface around Γ is reshaped as shown in Fig. 3B and as noted by the referee, which yields a broadened QWS fingerprint in the calculated STS spectrum (see below, 5Pb @ 3 BP) close to E_F .

Since this feature is not visible in the experimental STS data for 5 ML Pb and since it is highly dependent on the exact initial positions of the Pb QWS as well as on the underlying BP band structure, we interpret this as a modeling artifact, which is why we did not discuss it in the first version of the manuscript.

Fig. R4. Unfolded band structures for 5ML (left) and 6ML (right) Pb on 3ML BP. These are plotted along high-symmetry points of the primitive Pb (111) unit cell. Dashed red lines in the band structures refer to the pristine (but strained) 5ML and 6ML Pb band structure. Blue dots correspond to the unfolded spectral function as defined in Eq.(4) in the supporting text S3, where the dot-size represents the intensity of $A(k,w)$ at the corresponding momentum and energy. The side panels show the quasi-local spectral functions $\rho(\omega)$ using smearing $\sigma_k=0.02$ Å. Blue lines represent unfolded heterostructure data, while the red and orange lines depict $\rho(\omega)$ for the pristine Pb and BP structures, respectively.

Finally, we note that the modifications to the outer Fermi surface areas (away from Γ) in a heterostructure consisting of 6ML Pb on 3 ML BP (which doesn't suffer from details of states around Γ and close to E_F) are very similar to those in 5 ML Pb on 3 ML BP:

Fig R5. DFT calculated Fermi surface for 6 ML Pb (left) and 6 ML Pb on 3 ML BP (right).

We thus argue that the BP-Pb hybridization with Pb states away from Γ matter most for the modifications to the SC properties of Pb.

We added corresponding remarks to the revised supplementary information on page #5.

I could not completely figure out what additional features in LDOS the authors are talking about in the stroke #190 and what peaks they attributed to QWS in Fig.3 c & d. Figure S8 clearly shows a peak at ~ 1.2 V for the 4 ML film, which shifts slightly to the left as the film thickness increases with an odd number of ML, which is consistent with QWS. The similar behavior I can guess for the 7ML, where the peak is beyond the energy range and is shifting to the left with increasing the thickness of the film. Do the authors have a wider energy range spectra for 3ML, 5ML and 7ML where this peak can be identified? Do these peaks have the same FWHH for the odd and even ML Pb films? Is it possible to add data for 8-9 ML films to the Fig. 3 c&d and mark the peaks corresponding with QWS.

We agree that understanding of the STS spectra is crucial for the interpretation of the BP-Pb hybridization at the interface. First, we like to note that in the full calculation considering the heterostructure of Pb and BP, we always find a bilayer oscillation [Fig. 3 C (now Fig. 4D) and S8 (now Fig. S10)]. Spectra with an even number of Pb layers look similar and so do those with an odd number of Pb layers. In the new Fig. 4, we visualize this by using blue colors for the odd thicknesses and red for the even ones. However, in contrast to the well-known bilayer oscillations of the QWS in Pb/Si(111), we find strongly broadened features, which show a bilayer dependence, which is more subtle due to the aforementioned broadening.

To understand the microscopic origin, we compare below the experimental STS data for 5 and 6 ML of Pb at BP with DFT calculations of 5 and 6 ML of Pb on 3 ML of BP. The solid blue lines in the DFT calculations show the full LDOS, while the red and orange dashed lines indicate the LDOS for the individual subsystems, i.e. Pb and BP, respectively. For both calculations we can immediately see that the full LDOS is not just the simple sum of the individual LDOS, as significant hybridization

between the two subsystems yields shifts and broadenings of the QWS and is responsible for charge transfer from Pb to BP.

Most importantly, we find the main theoretical STS features in the experimental STS, although their exact positions are not perfectly inline due to approximations within the calculations.

For the 5 ML Pb system, we find:

- below -1eV an increasing shoulder, which we identify with BP states hybridized with a low-lying Pb QWS
- a gap of about 500meV around 0.5eV, which results from a lack of states in this window around Gamma
- a wide shoulder starting slightly above 0eV and extending to about 1eV showing a small feature around 0eV; we identify this with the initial QWS which overlaps (towards higher energies) with BP states; note that in our calculations the BP substrate gets effectively n -doped due to charge transfer from Pb to BP, thus the Pb QWS state here is hybridizing with the BP valence states

For the 6 ML Pb system, we find:

- an extended rather flat feature between -0.5 and 0.25eV, which we can identify with a strongly hybridized and renormalized Pb QWS states, which is shifted to higher energies due to the interaction with the BP states
- a feature with a local maximum at around 1eV (this is the feature referred to as “*peak at ~1.2 V for the 4 ML film*” by the referee). This results from a Pb QWS state on the background of an BP band as also seen in 5ML Pb, but without the overlaying QWS state.

From this analysis we understand that we indeed see a variety of Pb QWS features in the STS data, which are however either on top of the constant BP background or hybridized with and renormalized by the latter. Since this BP substrate induced renormalization and hybridization from the interface is less effective in the thicker films (i.e. >28ML as shown in Fig. S8) the resulting STS data is more reminiscent of the well-known Pb/Si(111) STS data.

In order to improve this discussion, we have split the previous Fig. 3 and added more information into the two figures (Fig. 3 and 4), and first expanded on the theoretical discussion in the manuscript to explain these points. The new Fig. 4, also details a decomposition of the band structure, which was previously in the supplemental information.

Fig. R6. Top panels show scanning tunneling spectra measured on 5ML (left) and 6ML (right) film. Bottom panels show the quasi-local spectral functions $\rho(\omega)$, where blue lines represent unfolded heterostructure data for 5ML Pb on 3ML BP (left) and 6ML Pb on 3ML BP (right). Orange and red and orange dashed lines in each panels shows $\rho(\omega)$ for corresponding pristine Pb and BP layers. Arrows in the figure shows correspondence between calculated and measured LDOS.

Could the broadening of QWS might be caused by the in homogeneity of the film, which has varying thickness and stronger relative variation for thin films than for films >30ML?

This is an interesting suggestion. However, we observe that the hybridized QWS are very reproducible, regardless of the film roughness (e.g. flat films or films with a certain number of islands/vacancy islands (See Fig. S8 and Fig. S10). We also observe similar spectra for individual islands grown on BP at room temperature (See Fig. S9). Using the fact that the islands (see Fig. S9) show similar spectra, we can unambiguously identify the island heights as there is no wetting later. Therefore, we would argue that the broadening effects cannot be explained by film inhomogeneity, but rather due to the interfacial hybridization as discussed above. **We have clarified this point on page 8.**

Reviewer #3 (Remarks to the Author):

In this manuscript the authors investigate the superconducting properties of thin Pb films grown on a highly anisotropic black phosphorus (BP) substrate. It is found that the superconducting gap and the vortex structure of the thin film undergo an anisotropic renormalization as a result of the coupling to the BP substrate. The presence of a strong moiré lattice and a uniaxial strain of the Pb film are also reported. While the results of this work are potentially interesting, I find the presentation rather confusing and I believe that the manuscript should not be published in this form. A few specific points are detailed below.

We thank the referee for this comment. In this revision, we have made major changes and added additional supplement information to clarify multiple points and to optimize the presentation.

The experimental observations reveal certain specific properties of this hybrid system and I believe that a key task is to understand the mechanism(s) responsible for the observed features. For example, the presence of the interface may induce electrostatic effects (band bending), strain (which, in turn, may affect the electronic properties), hybridization effects (which involves mixing of the electronic wave functions associated with the two subsystems), and various spin-dependent effects. The picture that emerges from the manuscript is rather murky. For example, it is suggested that, as a result of hybridization at the interface, the wave function in thin Pb films becomes delocalized. I assume that this implies that the electrons penetrate into the BP. If this is the case, how does one explain the absence of a proximity induced gap in the BP (at least near the interface)? What are the specific roles of electrostatic effects, strain, and hybridization? Note that the title suggests that the superconducting properties can be “tuned” using the hybridization at the interface. In this context, one should explicitly check that the observed properties cannot be generated by, e.g., a strained Pb lattice (i.e., without invoking the electronic properties of BP).

We thank the referee for this question. We agree with the referee that there are multiple degrees of freedom that in turn may affect the measured electronic structure and superconductivity. Below, we address the raised points individually:

Band bending/proximity effect: this question has been partially answered in reply to referee 1. Pb makes the surface of BP more insulating. Pb *n*-dopes the BP surface, which is natively *p*-doped, which we confirmed using STS. We detail this in revised and newly added Fig. S13. In these figures, we show islands of Pb grown where bare BP is exposed. One can see that there is a hard gap at low voltage (5 – 11 meV), which hampers the ability to tunnel through the bulk of the sample. Likewise, the apparent valence band edges of the BP are shifted, due to the apparent *n*-doping from Pb. This implies that there is an accumulation layer near the BP/Pb interface, which creates a weak insulating,

Schottky-like barrier. This “upwards band bending” rules out that there is a 2DEG formation at the interface, and explains why there is no observed proximitized superconductivity in the BP. As a result of this band bending, we have to work with closed Pb films that are contacted at the surface, in order to ensure the proper conductivity to measure the superconducting gap. We can also confirm this behavior in “holes” of certain closed films, which penetrate to the interface. In these “holes” we see no residual proximity effect, due to the insulating behavior (i.e. the upwards band bending).

Hybridization: the renormalized superconductivity in this paper is associated with hybridization. The most direct evidence of hybridization is the rather strong changes that are made to the QWS of the Pb film. In the limit of hard-wall potentials, namely the interface between wide-gap semiconductor/film and film/vacuum, the confinement can be described in a simple 2D potential picture. For Pb this is prototypical when grown on semiconductors, like Si(111), where the QWS near the Γ point show weakly dispersing flat like bands. In STS, this appears as sharp peaks at given energies, and shows a strong bilayer oscillation (see discussion above and, e.g., see ref PRL 96, 027005 (2006)). In contrast, over large energy windows we only see weak signatures of QWS with strong broadening, which points towards more dispersing states. From comparisons to DFT and model calculations, we conclude this is clear evidence of hybridization of the Pb bands with the BP bands. Moreover, this strong renormalization destroys the typical quantum size effects seen for Pb on other semiconductors, where bilayer islands are typically observed. Based on these large energy range observations of hybridization, it is evident that the Fermi surface will be strongly modified, which in turn leads to the weighting behavior we see and describe in Fig. 4 (Fig. 5). For the future, it will be of interest to understand how these modifications also modify the electron-phonon coupling and how this might affect the measured variations we see in Δ .

Strain: From our analytic model we understand that we need (a) an anisotropic Fermi surface yielding the weighting function $w(\theta)=1/F(\theta)$ and (b) an anisotropic gap function $\Delta(\theta)$, which (c) need to be in phase, i.e., $w(\theta)$ and $\Delta(\theta)$ are minimized / maximized for the same θ , to precisely reproduce the characteristics of the measured gap spectral function, c.f. Fig. 4 (Fig. 5) in the manuscript.

While the uniaxial strain doubtlessly introduces some $w(\theta)$ with reduced two-fold symmetry it must not necessarily lead to an anisotropic gap function $\Delta(\theta)$. Without the influence of the BP substrate, the Pb intrinsic gap function can get anisotropic only in case the strain would yield a significantly anisotropic electron-phonon coupling. Whether this happens and simultaneously results in in-phase $\Delta(\theta)$ and $w(\theta)$ can a priori not be said and requires very careful SC *ab initio* calculations for thin Pb films.

We, however, need to stress that our suggested BP-Pb hybridization mechanism (a) automatically yields the “phase-locking” of $\Delta(\theta)$ and $w(\theta)$ and (b) does not rely on supposedly introduced anisotropic electron-phonon couplings. In our proposal the anisotropic $\Delta(\theta)$ results from the new intrinsically anisotropic gap equation (15) from the Supplemental Material. Together with the clear evidence of BP-Pb hybridization from our STS measurements of the QWS, we can thus state that our proposed mechanism is certainly present and active but might be further refined by strain.

Spin: While the spin degree of freedom does not play an important role for BP, it has indeed been shown that spin-orbit coupling can influence the detailed characteristics of the QWS in free standing Pb films [see, e.g., PRB 87, 085440 (2013)]. The Fermi surface away from the Γ point is, however, just weakly affected. We thus expect spin-orbit coupling effects to play just a quantitative but no qualitative role here.

In the revised manuscript, we have added a discussion on page 11 to clarify these points.

In addition, I think that some of the language should be more transparent (and exact). For example, what does it mean that the “hybrid electronic structure derived at the interface ... can be activated to tune the superconductivity in the lead film.”?

We have modified the phrasing in the introduction, to clarify this point.

In general, what is the meaning of “tuning” in this context? Also, what should one understand by the concept of “interface electronic structure”. Note that the experimental results concern the electronic properties of a thin Pb film, which are perturbed by the PB substrate. I do not exactly see in what sense one can view the “interfacial electronic structure” as “a new degree of freedom”. I would be much more comfortable to simply say that the effects of Pb-BP hybridization depend on the (Pb) film thickness.

We agree that this is not very well clarified in the manuscript. One of the main claims of our paper is the superconductivity in the Pb film is modulated by the hybridization between the two materials. Hybridization in this limit does not quench superconductivity, but selectively modulates the superconducting quasiparticles in k -space. On a larger energy scale, this leads to a “leaky” quantum well that modulates the confinement potential on larger energy scales. In this way, the hybridization leads to a renormalization which can be captured with the weighting function together with the augmented gap equation we describe in our two-band model.

The resultant renormalization can be ascribed to many effects, as mentioned by the referee above, and may be tuned. For example, as detailed above, sample-to-sample variations in the band offsets of the BP leads to an observed modulation in superconductivity between different prepared films. The detailed hybridization between the two materials depends strongly on the relative band offsets, and modifying (i.e. gating) this could potentially be tuned in an exfoliated structure. Moreover, the gap renormalization is linked to the anisotropic band structure of BP which imprints itself on the Fermi surface of Pb. This reshaping of the Fermi surface is linked to the interplay between the two-fold and three-fold symmetry of BP and Pb, respectively (i.e. leads to the moiré lattice). While this is not explicitly shown in our paper, a change in the orientation of the two materials with respect to each other (e.g. via twisting) will lead to different hybridization conditions. This will in turn affect the Fermi surface, changing the weighting and the augmented gap equation and thus modulating the superconductivity. Such an idea should hypothetically have no effect on a typical isotropic BCS thin film superconductor, as the superconducting quasiparticles are isotropic in k -space. These are a few examples of parameters that when changed may modulate the superconductivity in the thin film. One potential material would ReSe₂ (see DOI: 10.1126/sciadv.aaw2347), which is suggested to also be an anisotropic semiconductor. While the material has a larger band gap, compared to BP, the material can be gating between both band edges. In proximity to a band edge, weak hybridization should kick in and the superconducting properties of the parent Pb film should be thus tunable. **We have added this discussion to the conclusion.**

With regards to the point about the interfacial electronic structure, we agree that this is vague and **have changed the wording in the manuscript.** We find it important to point out that Pb thin films on BP can no longer be thought about in one regime dictated solely by the thickness of the Pb film (i.e. the quantum confinement). In this work, we can argue there are at least two regimes, resulting from the interplay of the film thickness and the effective hybridization at the interface. We observed that thin Pb films, e.g. on the order of >30ML, are type II superconductors with QWS that are reminiscent to 30ML films grown on Si(111). However, ultra-thin films show a stronger renormalization and anisotropic behavior. We agree that the effect of the interface, both electronically and structurally will be reduced with increasing film thickness. In future studies, it will be interesting to understand how the interface affects in detail the role of electron-phonon coupling and spin-orbit coupling.

The claim that the measured dI/dV features in Fig. 3B “compare well to the calculated DOS (Fig. 3D)” requires further explanation (since the correspondence is not obvious).

We agree that this discussion was poor in the previous version of the manuscript. We have improved **this discussion on page 8**, including updating the figures.

REVIEWER COMMENTS

Reviewer #1 (Remarks to the Author):

I carefully read the revised manuscript and the reply. However, I still cannot be convinced by the data that the reduced symmetry of superconductivity exists. I cannot recommend the publication of the manuscript in Nature Communications. The following comments may be helpful for the authors if they want to improve their manuscript and submit it to another journal.

1. The SC gap data here can only serve as complimentary evidence since it reflects the integrated gap function over the whole BZ. The authors need to provide momentum resolved data, such as QPI results?
2. I don't see strongly anisotropic vortices in Fig. 2B. Some vortices show moderate elongation but along multiple (different) directions. If the hybridization of the electronic states with substrate exists, we should expect the same orientation of the elongation. I would use 'irregular vortices', since they are in random shape and elongated along different directions.
3. How do the authors find the vortex core in Fig. 2C? I understand that the core of an irregular vortex is unnecessarily located at the geometric center. But I don't see any reason to determine the vortex core at the crossing point of line 1 and 2. When I look into the line-cut 2 in Fig. 2D, I find all the spectra have a two-peak feature and never merge into one. It indicates that the Line-cut 2 probably deviates from the vortex core. In this case, there's no surprise for the discrepancies between the Line-cut 1 and 2.
4. To address SC anisotropy in a more quantitative level, the author should analyze the coherence length along the different directions by fitting the ZBC in various vortices. I find some fitting in Fig. S7. But those radially averaged data lose the information of the symmetry.
5. I can't clearly see any vortex in Fig S6. If the crossing point of line1 and line 2 is a vortex core, why the ZBC at that crossing point is lower than the nearby regions? I feel confused the data quality of the vortex in 30 ML Pb is lower than thinner films.
6. The QWSs in STS of Fig. 3D are inconsistent with the calculation results in Fig. 3C. For example, the ii peak in 6 ML has at least 0.5 eV energy deviation to the calculated ii peak. Such deviation is comparable to the hybridization induced energy shift of this peak shown in calculation result (Fig. 3B). This discrepancy makes the band hybridization less convincing. Other peaks also have large deviations.
7. The most significant symmetry breaking in the manuscript is the Moire pattern in topographic image. However, the symmetry breaking of band structure is less convincing and at least too tiny.

Reviewer #2 (Remarks to the Author):

The authors have addressed all of my technical comments reasonably well and have made corrections to the manuscript, that made a clearer focus on the main results, as well as easy to follow the argumentation and analysis of the experimental data. As I mentioned earlier, the presented results are of a high level, certainly have novelty and relevance for the study of the effect of interface on the electronic properties in ultrathin superconducting films. I can recommend articles for publication in this form. I thank the authors for their work.

Reviewer #3 (Remarks to the Author):

In this resubmission the authors address the issue raised in the previous round quite thoroughly. I find their answers satisfactory and I believe that the changes in the revised manuscript are useful in clarifying most of the ambiguities present in the previous version. Also, there is new relevant information (e.g., Figs. S4 and S13 and the corresponding discussion in the main text). It is not absolutely clear to me whether or not this work will have a significant impact on future research, but it suggests an interesting direction. I think that the manuscript contains enough elements to be considered for publication.

Reviewer #1 (Remarks to the Author):

Below, we answer the reviewer's comments, point by point. However, we do not see any reply to our previous rebuttal, and at certain points unjustified criticisms that appear contradictory. We also see no attempt to provide alternative explanations, neglected the mounting scientific information provided in the manuscript. In our previous reply, we elaborated on the gap renormalization (v -shaping), as well as on the QWS spectra (large energy scale spectra), which is reiterated here without acknowledgement of statements we previously made in our extended rebuttal (as well as to the other reviewers). As the reviewer does not provide any rebuttal to our previous reply, in the statements below, we assume the reviewer agrees with our previous rebuttal, and now brings up new concerns that were previously unmentioned. We below rebut all scientific points. Most points are already addressed in the manuscript, or in our previous rebuttal. Where appropriate, we now introduce additional clarification in the manuscript (highlighted).

I carefully read the revised manuscript and the reply. However, I still cannot be convinced by the data that the reduced symmetry of superconductivity exists. I cannot recommend the publication of the manuscript in Nature Communications. The following comments may be helpful for the authors if they want to improve their manuscript and submit it to another journal.

1. The SC gap data here can only serve as complimentary evidence since it reflects the integrated gap function over the whole BZ. The authors need to provide momentum resolved data, such as QPI results?

What the reviewer proposes is most likely not experimentally feasible, and would constitute a new study. We agree that measuring QPI as a function of energy in the gap can provide momentum-resolved information, or more specifically q -dependent information. This practice is often used for unconventional superconductors, where the gap size and dispersion are much larger, compared to what we study here. These studies provide information about the quasiparticles in the gap, and information about k -space dependency.

The proposal to perform QPI measurements here is, however, unfeasible and not practical, for the following reasons:

(1) QPI measurements require scattering centers (i.e. unwanted impurities), which are often present in higher concentrations in the aforementioned materials. This is at odds with the system we are studying here. For Pb on BP, we are in the clean limit, where the impurity levels are far too low to perform sufficient QPI measurements. In order to perform QPI, we would need to create dirtier materials (e.g. intentionally dope the material), which may inherently introduce disorder and complicate the interpretation. All of this is also predicated on the assumption that such impurities strongly scatter the quasiparticles, which is not always true. For example, we see no clear QPI signatures from step edges in the material.

(2) We are unaware of any study on thin film elemental superconductors, like Pb, where such QPI measurements have ever been made. In other words, this may be a standard practice in unconventional superconductors, but not for elemental superconductors. In order to measure any kind of dispersive behavior, QPI needs to be detectable over a significant energy range (i.e. measure $q(E)$). This requires significant quasiparticle densities in the gap, in addition to sufficient scatterers, as mentioned in (1). For unconventional superconductors, the superconducting gap is v -shaped and includes contributions of quasiparticles over large ranges of energy, in comparison to traditional BCS

superconductors. In our study, the gap is hard (i.e. there is a full gap over most of the energy range). Therefore, if we understand correctly, the reviewer is proposing that we study $q(E)$ over an energy range of the coherence peaks/conductance onset. That energy range is on the order of $\sim 0.2\text{meV}$, which is far too small to extract any meaningful dispersion, should there be any QPI signal and significant scatterers. Information about maps at those energies was provided already in reply to the previous rebuttal to another reviewer, with reference to maps in magnetic field and in reference to vortices. Therefore, this proposal to perform such a measurement, which has not been used to study these types of systems, is not appropriate, and most likely will not provide any new insight.

The absence of QPI-based experiments does furthermore not exclude the presence of an anisotropic gap. Although, the measured SC gap is integrated over the BZ, we have shown comprehensively in the manuscript that the anisotropic gap structure is due to non-thermal broadening (as we argued in our previous rebuttal). There is a thickness dependence and a renormalization of the gap structure (i.e. for ultra-thin films $<10\text{ML}$, and for films that are thicker), concomitant with quantified deviations in expected isotropic vortex behavior, as well as hybridized QWS, as we have previously rebutted. We have provided evidence based on numerous experimental and theoretical studies that the resultant gap renormalization is induced by the interface. Likewise, as stated in the manuscript and described in our previous rebuttal: the coherence length as well as the SC gap strength are different, in comparison to materials like Pb on Si. This refutes the statement that we can neglect the interface, as the experimental evidence cannot be substantiated by solely considering Pb subsystem itself. Finally, we stress: Pb films on Si(111), typically show a bilayer oscillation in the SC gap concomitant with magic island thicknesses (see for few of many more refs. *Science* **306**, 1915 (2004), *Phys. Rev. Lett.* **96**, 027005 (2006), *Phys. Rev. Lett.* **86**, 5116 (2001), *Phys. Rev. B* **65**, 245401 (2002)), as also previously addressed. This is quenched in the thin film limit here, enabling the formation of the Abrikosov lattice, all providing substantial evidence of the role of the interface. In this picture, our theoretical modeling and calculations show clear evidence of significant hybridization, which can influence the gap structure. As the reviewer also does not appeal the observation of this gap renormalization (as we have previously rebutted), we do not see here any substance for an alternative explanation, which could explain our experimental data in such detail as already provided.

2. I don't see strongly anisotropic vortices in Fig. 2B. Some vortices show moderate elongation but along multiple (different) directions. If the hybridization of the electronic states with substrate exists, we should expect the same orientation of the elongation. I would use 'irregular vortices', since they are in random shape and elongated along different directions.

We do not fully understand the reviewer's concerns, but believe that part of the comment refers to semantics we find misleading, and the other to a scientific question about symmetries. Concerning semantics: vortices that "*show moderate elongation*" are not isotropic, i.e., they are anisotropic. For reference, please see the following references for vortex structures of Pb on other substrates: *Phys. Rev. Lett.* **101**, 167001 (2008), *Nature Physics* **6**, 104-108 (2010) and *APL* **103**, 242603 (2013). In these references, the vortices of thin Pb films on silicon are isotropic. As the reviewer acknowledges, the vortices are elongated and do not exhibit a radial symmetry, confirmed by Fig. S5. Therefore, the use of the word anisotropic is appropriate.

We believe the use of the word "*irregular*" may create a misconception that the superconductor is strongly disordered. However, this is not accurate: we see a clear Abrikosov lattice. It is important to note that Abrikosov lattices have not regularly been observed in Pb thin films before, due to the

strong influence of the QWS on superconductivity for films with multiple thicknesses. *This in itself is a new observation: as the interfacial hybridization has a strong influence in the ultra-thin film limit, the gap is nearly thickness independent for <10ML.* That allows for the formation of the Abrikosov lattice over a large length scale, in contrast to previous measurements on larger band gap semiconductors.

We do agree with the referee that it is difficult to identify a particular symmetry/orientation consistent for all vortices (e.g. three-fold or two-fold). This interesting question may be the subject of future study. However, there is a possible scientific explanation for this based on hybrid superconductivity: If the superconductivity is driven by the interface, then we expect a vortex structure to exhibit a symmetry of the given Fermi surface. For Pb, this is three-fold symmetric, while for BP, this is two-fold symmetric. In addition, there is a long-range moiré structure, and with these symmetry considerations, it is natural to expect a reduced symmetry. It is established that potential modulations may influence the shape of a vortex, as well as create pinning centers. For reference, we include a figure of the topography of the sample below (Fig. R1), for the vortices measured in Fig. 2. **We have also added this topography in the supplementary Figure S5.** The film is far from perfectly flat, and it was in fact surprising to see an Abrikosov lattice at all. Therefore, we expect all these factors to lead to vortices with a reduced symmetry that exhibit local variations. We have made one other scientific observation: most vortices exhibit “ray” like structures, reminiscent of those seen on NbSe₂, and these are oriented along certain crystallographic directions. This can also be clearly seen in Fig. S7, where there is evidence of certain orientations consistent among vortices. As there are already sentences detailing this, we argue this is explained in the manuscript. In addition, the shape of the vortex varies in shape and size that could be due to the weak electronic disorder due to LDOS variations resulting from the film thickness and the moiré pattern (see page 6). **We have modified the text in the conclusion, where this was mentioned, and elaborated more.**

Figure R1: Constant current STM image of the Pb film showing area for vortex imaging, in the same area as Fig. 2B of the main manuscript and Fig. S5 of the supplement. Imaging parameters $V = 10$ mV, $I = 10$ pA (scale bar = 200 nm).

3. How do the authors find the vortex core in Fig. 2C? I understand that the core of an irregular vortex is unnecessarily located at the geometric center. But I don't see any reason to determine the vortex core at the crossing point of line 1 and 2. When I look into the line-cut 2 in Fig. 2D, I find all the spectra have a two-peak feature and never merge into one. It indicates that the Line-cut 2 probably deviates from the vortex core. In this case, there's no surprise for the discrepancies between the Line-cut 1 and 2.

We define the core of the vortex as the point where there is an observed peak at zero energy, in the STS (for reference see, e.g., PRL, **62**, 214 (1989)). Comparing images in Fig. S5 showing the ZBC map as well as the map at the coherence peak, the vortex core is at the geometric center of the anisotropic features seen in the ZBC map. As we previously mentioned in our reply to the reviewer, the spectra were measured along the lines to avoid island edges and line-2 appears to be a few nm shifted from the geometric center, However, from the line-1 (which passes through the geometric center), it is clear that spectral features in the gap vary over larger length scale than the shift in the line-2 from the geometric center. We therefore disagree with the reviewers' comment on the observed discrepancies between the spectral features in line-1 and 2 as from the simple geometric arguments one would expect the two peaks in line-2 would merge into coherence peaks in a same way as line-1 and also over the smaller length scale. As mentioned in the previous rebuttal, the evolution of spectral features in different directions is scientifically accepted evidence for an anisotropic vortex core (see PRL, **62**, 214 (1989), PRB **56**, 9052 (1997), PRB **53**, 15316 (1996)).

4. To address SC anisotropy in a more quantitative level, the author should analyze the coherence length along the different directions by fitting the ZBC in various vortices. I find some fitting in Fig. S7. But those radially averaged data lose the information of the symmetry.

As per the reviewer's suggestion we analyzed the evolution of the ZBC profile through two vortices along two directions, namely: (1) along one of the high symmetry axes with bright rays (profile 1 and 3) and (2) along the direction 30° rotated away from the high symmetry axes where ZBC shows

Figure R2: Coherence length calculations from line profiles. (A) Zero bias dI/dV map at $B_{\perp} = 50$ mT, showing an Abrikosov vortex lattice (same as Fig. 2B of the main manuscript). Imaging parameters for (B) and (C): $V_{stab} = 10$ mV, $I_{stab} = 10$ pA, $V_{mod} = 200$ μ V, $\Delta z = -80$ pm, scale bar = 100 nm. (B-E) Line profiles across two vortices marked in (A) together with the exponential fit using a least squares method.

reduced intensities (profile 2 and 4). Exponential fitting to these profiles shows that the coherence length is significantly different in the two directions, further ascertaining the claim of anisotropic vortices. Nevertheless, further study is needed to acquire adequate statistics to quantify the coherence length as a function of orientation, due to the variations between observed vortices. Such an analysis would make more sense on flatter films, where more uniformity may be expected

between vortices. We have added a sentence to the manuscript to suggest that the fitted coherence lengths in different directions yield further values but require further study.

5. I can't clearly see any vortex in Fig S6. If the crossing point of line1 and line 2 is a vortex core, why the ZBC at that crossing point is lower than the nearby regions? I feel confused the data quality of the vortex in 30 ML Pb is lower than thinner films.

We would like draw attention to the main point of Fig. S6: as the film gets thicker, (a) the gap renormalization is reduced (see Fig. 2), (b) the quantum well states start to resemble Pb films on Si(111), where a bilayer oscillation as well as peak structures can be identified (see Fig. S11), and (c) the vortex characteristics are *different* than what is seen for thinner films (e.g. <10 ML). These three points provide very strong evidence of our main claim: interfacial hybridization drives superconductivity in the ultra-thin film limit. We do not convey that the vortex behavior seen in the range of 30ML, as shown in Fig. S6, is expected to be the same as that in the ultra-thin film limit (<10ML, e.g. seen in Fig. 2). This precise difference, is another piece of evidence of the role of the interface. As the film gets thicker, and the structural and electronic effects of the interface are screened, the interfacial contribution to the superconductivity and electronic structure (i.e. QWS) is strongly reduced. All three observations are consistent with this claim.

Concerning vortex maps in thicker films: we only observe a “bright” feature in spectroscopy and not in the given maps. This is a result of the measurement method, i.e., to acquire ZBC maps the tip is stabilized at higher energy and becomes more susceptible to local variations in the LDOS. At higher thicknesses, there are significantly more variations in the signal due to the increased number of QWS, as well as the variations in thickness, making it difficult to “see” the vortex core. Nevertheless, the peak structure corresponding to the vortex core can be seen in point spectroscopy taken along a line: see Fig. S6C. *Most importantly, the line spectra taken along the vortex in different directions is much more isotropic in comparison to that of the spectra shown in Fig. 2C for films in the ultra-thin limit. This is another piece of evidence that the role of the interface is weaker as the film is thicker.*

6. The QWSs in STS of Fig. 3D are inconsistent with the calculation results in Fig. 3C. For example, the ii peak in 6 ML has at least 0.5 eV energy deviation to the calculated ii peak. Such deviation is comparable to the hybridization induced energy shift of this peak shown in calculation result (Fig. 3B). This discrepancy makes the band hybridization less convincing. Other peaks also have large deviations.

The reviewer indicates that the discrepancy between the precise positions of the DFT calculations with the experimental data renders the interpretation of band hybridization less convincing. We do not agree with this point, and there is sufficient and non-negligible experimental evidence for this statement, independent of the QWS calculations, as we also rebutted in our previous reply concerning the QWS spectra. For comparison, we advise the reviewer to see the QWS in *Phys. Rev. Lett.* **96**, 027005 (2006) for Pb on Si, where clear peak structures due to the QWS can be identified with a relatively featureless background This can be directly compared to the QWS spectra shown in the manuscript as well as highlighted in the previous rebuttal. It can be unambiguously seen that the apparent peaks seen in STS for Pb on BP are strongly broadened, and include many other additional features, which the reviewer previously identified as noise. *These modifications cannot be neglected.* As to the latter point, this was rebutted in our previous reply, where we showed that these are features of the electronic structure. These observations are significant deviations from the electronic

structure of the QWS of Pb in the nearly-free-standing limit (e.g. for Pb/Si). As the only other interface is vacuum, the natural conclusion is that these features (e.g. broadening of peaks, and additional features), are due to the interface. As these features become diminished, as the film gets thicker (e.g. compare 30ML range to <10ML), the most direct conclusion is interfacial hybridization.

Finally, as we detailed in the manuscript, the quantum well state influence on the growth of Pb films/islands is quenched (e.g. see ref. *Phys. Rev. Lett.* **86**, 5116 (2001), *Phys. Rev. B* 65, 245401 (2002)). This latter point is a clear indication that the expected surface energy oscillations are drastically reduced, resulting from a change in the electronic structure. For Pb on materials like Si(111), the presence of a larger band-gap would imply that the interface can be considered as a hard wall potential in a large energy range, leading to quantum confinement. As BP is a narrow band-gap semiconductor, the conclusion is that the confinement barrier at the interface is reduced, and due to hybridization, there are electronic states of BP that weaken the confinement potential. This point is sufficiently supported by *ab initio* calculations, which clearly show that at many energies, there is hybridization between the BP and Pb bands. *We remind the reviewer: QWS can be seen for Pb on Si, and other materials over an energy range of nearly 2 eV. For a material like BP, with a band gap of 0.3 eV, bulk states cannot be ignored, regardless of the relative band offsets of the two materials within a reasonable range.* These points were also detailed in our previous rebuttal, and better exemplified by the modified Figure 3. We believe this point is adequately addressed in the current version of the manuscript.

The precise electronic structure and resultant hybridization between BP/Pb bands depends strongly on the absolute band offsets between the different materials. *We want to make clear: calculating the band structure of the exact film is not computationally feasible.* This was detailed in our previous rebuttal in response to the other reviewers, which can be referenced for more details. In order to provide further clarity: we do not claim that the calculations are an exact representation of our experimental data, as the calculations consider (a) three ML of substrate and not its bulk form, (b) a reduced moiré cell, and (c) do not take substrate doping into account as in the experiments. Therefore, the calculations only *qualitatively* explain the impact of hybridization on the electronic structure of the Pb film, but as adequately as possible. Our calculations clearly show that there are qualitative features that are comparable and explain at given energy ranges where the QWS of Pb overlap with BP bands, and that *there is significant hybridization*. This leads to an effective broadening of the QWS as well as an anisotropic dispersion of those states, *which is not seen in free-standing lead*. Our calculations show that there are a number of energetic locations, where any potential Pb QWS may hybridize with the bands of BP. As the energetic location of the QWS of free standing Pb oscillate as a function of thickness, and the difference in energy of these states decreases with increasing thickness, this qualitative picture that hybridization renormalizes the electronic states of the Pb is justified.

7. The most significant symmetry breaking in the manuscript is the Moire pattern in topographic image. However, the symmetry breaking of band structure is less convincing and at least too tiny.

We do not understand the reviewer's comment, and refute this statement concerning the tiny symmetry breaking effects stemming from the band structure. We refer to answer 6 above. The use of the word "tiny" directly suggests that the effects of the hybrid band structure are negligible, i.e., that the influence of the BP substrate is negligible. This is refuted by the overwhelming amount of information provided. In addition to all the information provided, both other reviewers disagree with the statement that the renormalization of the QWS is tiny. Likewise, in our previous rebuttal,

we addressed the comment from the reviewer why there are additional features in the STS in a large energy range, and that this is not attributed to noise. This indicates that such experimental observations are not tiny, i.e. negligible, and deviate from spectra taken on nearly free-standing Pb films. Furthermore, should the renormalization of the band structure (e.g. Fig. 3,4, S9-11) be tiny, then we believe the reviewer is suggesting that the experimental/theoretical data should be comparable to a free-standing film where hybridization is neglected. In other words, experimentally and theoretically we should see the same qualitative LDOS seen for numerous other Pb/substrate combinations and that the spectra shown in figure provided above are identical. We believe this is substantially refuted.